# Allele-specific transcriptional effects of subclonal copy number alterations enable genotype-phenotype mapping in cancer cells

Hongyu Shi[1,2], Marc J. Williams [1], Gryte Satas[1], Adam C. Weiner [1,3], Andrew McPherson [1] & Sohrab P. Shah [1] ✉

Subclonal copy number alterations are a prevalent feature in tumors with high chromosomal instability and result in heterogeneous cancer cell populations with distinct phenotypes. However, the extent to which subclonal copy number alterations contribute to clone-specific phenotypes remains poorly understood. We develop TreeAlign, which computationally integrates independently sampled single-cell DNA and RNA sequencing data from the same cell population. TreeAlign accurately encodes dosage effects from subclonal copy number alterations, the impact of allelic imbalance on allele-specific transcription, and obviates the need to define genotypic clones from a phylogeny a priori, leading to highly granular definitions of clones with distinct expression programs. These improvements enable clone-clone gene expression comparisons with higher resolution and identification of expression programs that are genomically independent. Our approach sets the stage for dissecting the relative contribution of fixed genomic alterations and dynamic epigenetic processes on gene expression programs in cancer.

Genomic instability is a hallmark of human cancer which leads to copy number alterations (CNAs) in cancer cell genomes, and extensive intratumor heterogeneity[1–3]. It is well established that CNAs of driver oncogenes and tumor suppressors are causal determinants that change the fitness of cancer cells[4,5], leading to clonal expansions, clone-clone variation[6] and tumor evolution. In addition to impacting specific genes, CNAs often span chromosome arms or whole chromosomes and therefore potentiate transcriptional impact across hundreds of genes with a single genomic event. Recent reports on the extent of cell-to-cell variation of CNAs in tumors (including in well understood oncogenes)[3] raise the critical question of how granular subpopulations are phenotypically impacted by subclonal CNAs. Importantly, phenotypic impact of subclonal CNAs can have both cell intrinsic effects and act as cell-extrinsic determinants of the tumor microenvironment[7], further illustrating the importance of dissecting how CNAs modulate phenotypic intra-tumor heterogeneity.

Previous studies using bulk sequencing techniques have investigated the association between clonal CNAs and gene expression[8–11]. The expression level of a gene can be influenced by copy-number dosage effects reflected by the significant positive correlation between gene expression and the underlying copy number (CN)[12]. However, gene dosage effects are not deterministic and may be subject to compensatory mechanisms, rendering the impact of CNAs on expression as highly variable across the genome. Transcriptional adaptive mechanisms[13] including epigenetic modifications and downstream transcriptional regulation, can modulate CN dosage effects[14–16], further obscuring the direct impact of gene dosage. For example, the expression of certain immune response pathways often exhibit both CNA-dependent and CNA-independent expression[8].

Theoretically, measuring single cell RNA and DNA data should elucidate how genotypes translate to phenotypes at single cell resolution. Technologies that sequence both RNA and DNA modalities co-

[1]Computational Oncology, Department of Epidemiology and Biostatistics, Memorial Sloan Kettering Cancer Center, New York, NY, USA. [2]Gerstner Sloan Kettering Graduate School of Biomedical Sciences, New York, NY, USA. [3]Tri-Institutional PhD Program in Computational Biology and Medicine, Weill Cornell Medicine, New York, NY, USA. ✉e-mail: shahs3@mskcc.org

registered in the same cell would be ideal for linking genomic alterations to transcriptional changes in tumor evolution. However, pioneering technologies[17,18] have had limited throughput, lower quality and are still not mature enough for large-scale profiling of cancer cells. Sequencing single cell RNA or DNA independently allows more cells to be profiled and reveals a more comprehensive view of the cell populations, but requires computational integration of the two data modalities.

Several computational methods have been proposed for joint analysis of single cell DNA and RNA data. CloneAlign[19] is a probabilistic framework to assign transcriptional profiles to genomic subclones based on the assumption that the expression level of a gene is proportional to its underlying copy number. More recent methods SCATrEx[20] and CCNMF[21] are also based on this assumption but use different methods to model the similarity between copy number profiles and gene expression patterns. However, these methods do not consider the possibility that transcriptional effects of copy number could be variable between genes and therefore lack the specificity to decipher genes that may be subject to dosage effects from those that are independent of CNAs. In addition, these methods either require using predefined subclones from scDNA data or specify the number of subclones as input which may propagate errors of uninformative subclones or may miss more granular gene dosage effects. More importantly, the revelation of phenotypic plasticity as a driver of genetically independent transcription in cancer cells[22–24] motivates the need to disentangle genetic from epigenetic mechanisms. No available methods directly model dosage effects of subclonal CNAs, which is critical to infer which genes are deterministically modulated by subclonal CNAs and which genes are independent of CNAs. Moreover, recent advances have illuminated the extent to which allele-specific copy number alterations can mark clonal haplotypes both in DNA-based[3] and RNA-based[25] single cell analysis,

illustrating both a methodological gap and analytical opportunity for integration.

In this study, we address the questions of how subclonal CNAs drive phenotypic divergence and evolution in cancer cells, and quantitatively encode (allele specific) CN dosage effects in this process. We present a Bayesian method, TreeAlign, to enumerate and define CNA-driven clone-specific phenotypes, and also a statistical framework to compare the transcriptional readouts of genomically defined clones. TreeAlign implements a Bayesian probabilistic model that maps gene expression profiles from scRNA to genomic subclones from scDNA which i) can refine subclone definition from single cell phylogenies through a recursive process suggested by transcriptional divergence, ii) explicitly models dosage effects of each gene and iii) models allele-specific CNAs to better resolve clonal mappings. Through extensive benchmarking, we demonstrate that TreeAlign outperforms alternative approaches in both clone assignment and gene dosage effect prediction. Applying TreeAlign to both primary tumors and cancer cell lines, we characterized the phenotypic differences between tumor subclones, investigated the contribution of subclonal CNAs to clone-specific gene expression patterns in cancer cells and common expression programs which are altered by subconal CNAs.

## Results

### TreeAlign: a probabilistic graphical model for clone assignment and dosage effect inference

We developed TreeAlign, a probabilistic graphical model which maps scRNA sequenced cells to scDNA-derived subclones. TreeAlign employs a recursive algorithm for delineating subclones within phylogenies constructed from scDNA data (Fig. 1a). The model jointly infers clone assignments, clone-specific CN dosage effects and (optionally), models clone-specific allelic transcriptional effects (Fig. 1b). The TreeAlign framework assumes a subset of genes with

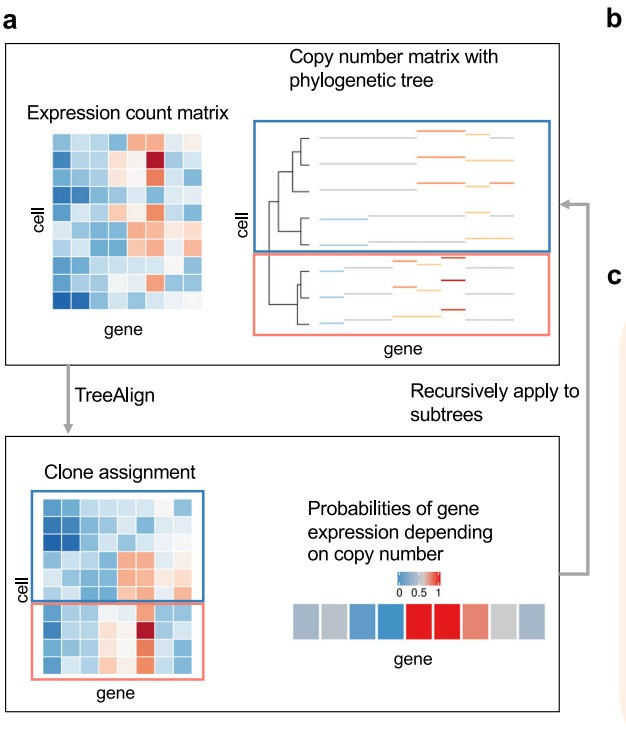

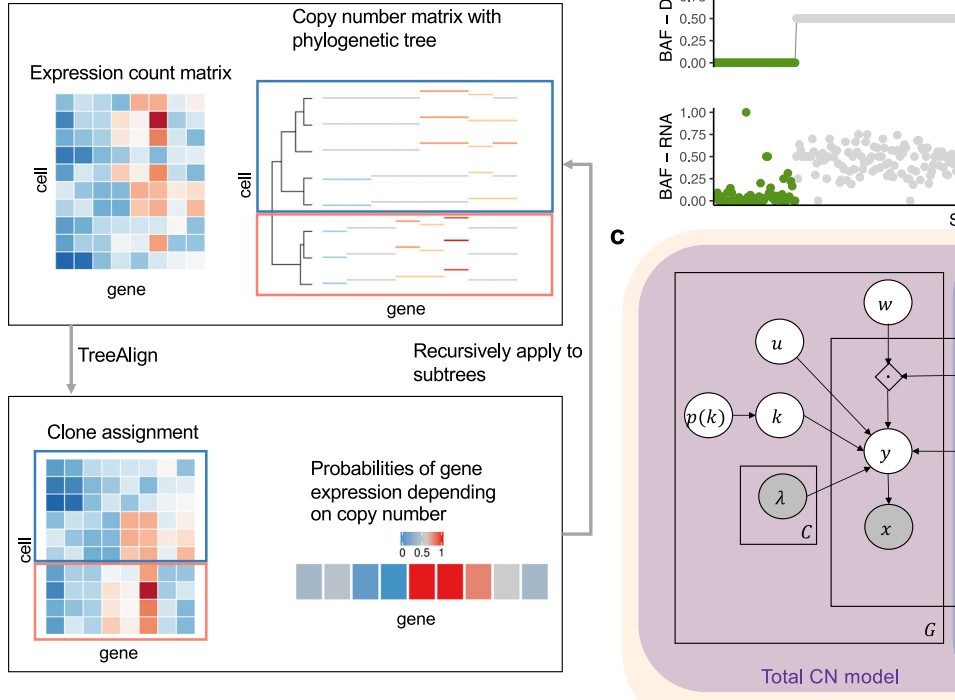

**Fig. 1 | Overview of TreeAlign. a** TreeAlign takes raw count data from scRNA-seq, the copy number matrix and the phylogenetic tree from scDNA-seq. By recursively assigning the expression profiles to phylogenetic subtrees, TreeAlign infers the clone-of-origin of cells identified in scRNA-seq and the dosage effects of subclonal

CNAs. **b** Allelic imbalance can be inferred from DNA data and RNA data. We assume a positive correlation between the two measurements to improve clone assignment. **c** Graphical model of TreeAlign.

positively correlated expressions to their underlying copy numbers. For each gene, expression is modeled by $k$, where $k \in \{0, 1\}$ is a Bernoulli variable such that the probability $p(k=1)$ represents the probability the gene has clone-specific CN dosage effects (Fig. 1c). This encoding allows us to frame the problem as a conditional probability distributon which separates the expected expression into two components. To infer clone assignments and $p(k)$, TreeAlign requires three inputs: (1) a cell × gene matrix of raw read counts from scRNA-seq, (2) a cell × gene copy number matrix estimated from scDNA data and (3) a phylogenetic tree (or optionally, predetermined clone labels) from scDNA profiles. TreeAlign can either assign expression profiles to predefined clone labels, similar to CloneAlign[19] or can operate on a phylogenetic tree directly to assign cells to clades of the phylogeny (Fig. 1a). When using a phylogenetic tree, a Bayesian hierarchical model is recursively applied starting from the root of the tree, computing the probability that expression profiles in scRNA can be mapped to a subtree. The stopping condition of the recursion is satisfied when the genomic or phenotypic differences between two subtrees become too small to allow confident assignment of expression profiles.

In addition to altered gene expression levels, allele-specific CNAs also lead to allele-specific expression imbalance which is detectable in scRNA data[3,26] (Fig. 1b). For example, genomic segments harboring loss of heterozygosity (LOH) deterministically leads to mono-allelic expression of genes in the segment while allelic imbalance owing to allele specific gains will skew the relative expression of specific alleles. To exploit how allelic imbalance modulates allele specific expression, we extended TreeAlign to model both total CN and allelic imbalance (Fig. 1c, Fig. S1). Given the B allele frequencies (BAFs) estimated from scDNA haplotype blocks using, for example, SIGNALS[3] and allele-specific expression at corresponding heterozygous SNPs in scRNA data, the allele-specific model contributes to clone assignment and infers the probability of the allele assignment $p(a=1)$, $a \in \{0, 1\}$, which indicates whether the SNP is on allele B or not. The total copy and allele-specific components of the probabilistic graphical model combine to form the 'integrated model'.

The software for all models of TreeAlign (https://github.com/shahcompbio/TreeAlign) is implemented in Python using Pyro[27] and is publicly available. Our implementation allows users to run the total CN model, allele-specific model and integrated model by providing different inputs. See "Methods" for additional mathematical, inference and implementation details.

## Performance of TreeAlign on simulated data

We first evaluated TreeAlign on synthetic datasets, quantifying the effect of three main parameters in the input data: number of cells (100–5000), number of genes (100–1000) and proportions of genes with dosage effects (10–100%). Simulations were performed using the generative model of CloneAlign[19]. We compared the performance of assigning expression profiles to ground truth predefined clones between TreeAlign, CloneAlign and InferCNV[28]. InferCNV was originally developed for inferring CNAs from gene expression data, but has also been repurposed for clone assignment in some studies[29]. InferCNV analysis in this context acts as a way of inferring clone assignment without the benefit of the scDNA data. Compared to CloneAlign and InferCNV, TreeAlign performed significantly better in terms of clone assignment accuracy especially in the regime where fewer genes exhibit dosage effects (Fig. 2a). For example, in the regime of 60% of genes with dosage effects (1000 cells, 500 genes), TreeAlign achieved mean clone assignment accuracy of 91.1%, compared to CloneAlign with 75.1% accuracy. The improvement in clone assignment accuracy was consistent across all cell and gene number simulation scenarios (Fig. S2). We also tested performance with phylogenetic tree inputs to evaluate if TreeAlign could achieve similar results on tree input compared to pre-defined clone input. Similar to the 'clone' regime, these simulations varied the proportion of genes with gene

dosage effects in 10% increments. TreeAlign was able to assign expression profiles back to the corresponding clades of the phylogeny with similar accuracies compared to the clone input in regimes with >40% genes with dosage effects (Fig. 2b, Fig. S3). Together these evaluations reflect that the model effectively obviates a priori tree cutting without paying a penalty in accurate clone mapping.

We also evaluated the accuracy of predicting dosage effects for each gene in the input datasets. We compared the simulated and predicted (using $p(k)$ as an estimate) frequency of genes with CN dosage effects. For high expression genes, simulated and predicted frequencies were highly concordant (Fig. 2c). For datasets with ≥50% of genes with dosage effects, the mean area under the receiver-operator curve (*AUC*) was ≥0.99 for genes with relatively high expression level (genes in top 40% in terms of normalized expression levels) (Fig. S4). We compared $p(k)$ to a baseline estimation of CN dosage effects which is the per-gene Pearson correlation coefficient (*R*) of CN and expression after fitting CloneAlign. $p(k)$ from TreeAlign had an overall higher *AUC* compared to *R* from CloneAlign for predicting CN dosage effects. This establishes that $p(k)$ captures gene dosage effects and has the ability to distinguish genes with dosage effects from those without dosage effects.

We then investigated how allele-specific information improves clone assignment with synthetic datasets. We simulated BAFs for varying numbers (0, 250, 500, 750 and 1000) of heterozygous SNPs with allelic-imbalance and simulated allele-specific expression from these SNPs using the generative model of allele-specific TreeAlign. We applied the integrated model on these synthetic datasets which contained total CN and allelic information, and confirmed that clone assignment accuracy was improved when more SNPs were included (Figs. S5 and S6).

## TreeAlign assigns HGSC expression profiles to phylogeny accurately

We next investigated TreeAlign's performance on real-world patient derived data from high grade serous ovarian cancer (HGSC). We first applied TreeAlign on single cell sequencing data from an HGSC patient (patient 022)[7]. Tumor samples were obtained from both left and right adnexa sites of the patient. scDNA ($n = 1050$ cells) and scRNA ($n = 4134$ cells) data were generated through Direct Library Preparation (DLP+)[30] and 10x genomics single-cell RNA-seq[31] respectively. 3579 (86.6%) ovarian cancer cells profiled by scRNA were assigned to 4 subclones identified by scDNA-seq. The expression profiles of clone C and D are overlapped on the UMAP embedding, while separated from the profiles of clone A and clone B, which coincides with the shorter phylogenetic distance between clone C and D (Fig. 3a). The separation of cells by assigned clones on the expression-based UMAP also suggests that the genetic subclones possess distinct transcriptional phenotypes.

We confirmed the clone assignment accuracy of TreeAlign by comparing the clonal frequencies estimated by RNA and DNA data (Fig. 3b). As both scRNA and scDNA data were generated by sampling from the same populations of cells, the clonal frequency estimated by the two methods should be consistent. Clonal frequencies in the left and right adnexa sample from the two modalities were significantly correlated ($R = 0.99$, $P = 9 \times 10^{-7}$). In addition, copy number alterations inferred for scRNA cells using InferCNV[28] were concordant with the scDNA based CNA of the clones to which scRNA cells were assigned (Fig. 3c). For example, notable clone specific copy number changes can be seen in both scDNA and scRNA on chromosome X in clone A. Clone B, C and D-specific amplification on 3q can also be observed in both scDNA and scRNA (Fig. 3d). By comparing the RNA-derived copy number profiles with scDNA data, we noticed that inferring copy number from RNA data is not always accurate. For example, the inferred profiles missed the focal amplification on chromosome 18. We also held out genes from chromosome 9 and chromosome 12 and re-

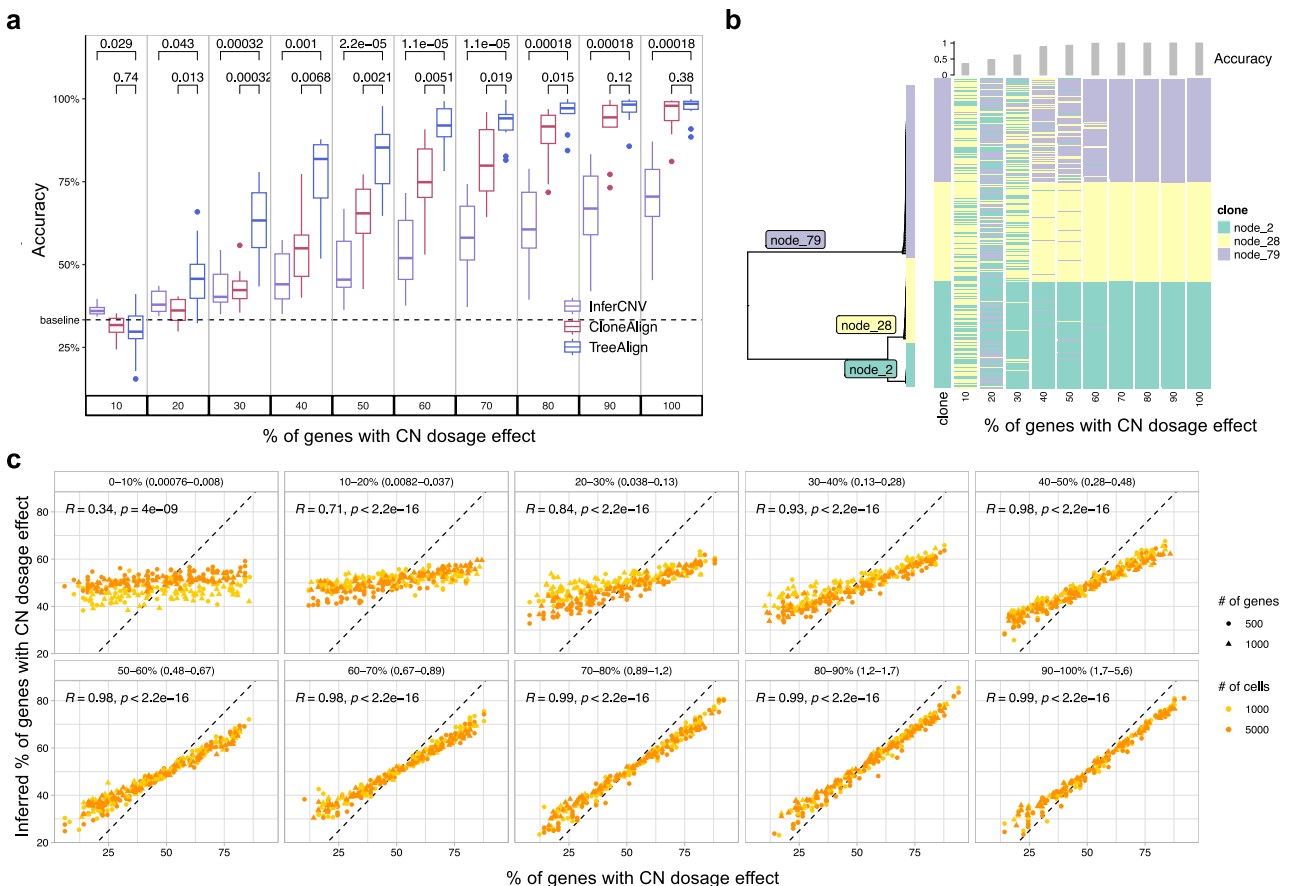

**Fig. 2 | Performance of TreeAlign on simulated data. a** Clone assignment accuracy of TreeAlign, CloneAlign and InferCNV on simulated datasets (500 cells, 1000 genes, 3 clones) containing varying proportions of genes with CN dosage effects. Brackets: *P* values with two-sided Wilcoxon signed-rank test. For the box plot, box limits extend from the 25th to 75th percentile, while the middle line represents the median. Whiskers extend to the largest value no further than 1.5 times the interquartile range (IQR) from each box hinge. Points beyond the whiskers are outliers. **b** Phylogenetic tree (left) of cells from patient 081 constructed using scDNA data. Heat map (right) of clone assignment by TreeAlign. Each column shows the assignment of simulated expression profiles to subtrees of the phylogeny. The bar chart above shows the overall accuracy of clone assignment. **c** Scatter plots comparing inferred gene dosage effect frequencies and the simulated frequencies. Each panel groups genes with similar expression levels from low expression genes (0–10%, with normalized expression between 0.00076–0.008) to high expression genes (90–100%, with normalized expression between 1.7 and 5.6). Pearson correlation coefficients (*R*) and P values for the linear fit (Two-sided Student's t-test) are shown. Source data are provided as a Source Data file.

ran TreeAlign with the remaining genes. 98.8% cells assigned by both the full and held-out dataset had consistently clone assignment labels. Clone level gene expression on chromosome 9 and 12 was consistent with the corresponding copy numbers (Fig. 3e). These results demonstrated a proof of principle that TreeAlign can properly integrate scRNA and scDNA datasets and highlighted that scDNA-seq can provide valuable information on CNAs and tumor subclonal structures which would be difficult to detect with expression data only.

We also applied TreeAlign to previously published data from a gastric cell line NCI-N87 generated by 10x genomics single-cell CNV and 10x scRNA assays[32]. TreeAlign assigned 3212 cells from scRNA to three clones identified in scDNA. The clonal frequencies estimated by both assays were closely aligned (Fig. S7). As for the patient 022 data, the scRNA cells showed subclonal copy number similar to the scDNA clones to which they were assigned, thus illustrating that TreeAlign also performs well on platforms other than DLP+.

**Incorporating allele specific expression increases clone assignment resolution**

We next investigated the extent to which accurate clone assignment solely based on allele specific expression could be performed. We inferred allele specific CN and BAF in scDNA data from patient 022 using SIGNALS[3]. The allele specific heat map (Fig. 4a) revealed

characteristic patterns of clonal LOH in whole chromosomes (e.g. chr 6, 13, 14, 17) as well as subclonal losses (e.g. chr 9q in clone A and parallel losses on chr 5 across multiple subclones). With the allele-specific model, we assigned cells from scRNA to clone A as identified by scDNA in patient 022. Clone assignments were consistent between the allele specific model and the total CN model with 87% cells concordant. The clone-specific frequencies of reads from B allele in scRNA accurately reflected scDNA BAF (Fig. S8a), with the exception of SNPs on chromosome X which showed allelic imbalance in scRNA but not in scDNA due to X-inactivation. These results suggest that allelic imbalance information can be effectively exploited for clonal mapping.

We then applied the integrated model utilizing both total CN and allele-specific information on data from patient 022. Relative to the total CN model, the integrated model mapped scRNA cells to smaller subclones (Fig. 4a). Specifically, we note when considering allele specificity, Clone B was subdivided into two subclones (B.1 and B.2). Clone B.1 had an additional deletion at 16q leading to LOH, whereas Clone B.2 had an amplification at 11q with increased BAF (Fig. 4a). Clone D was further divided into four subclones (D.1, D.2, D.3 and D.4). Clone D.1 and clone D.2 both had a deletion on chromosome 5, but the deletion events occurred on different alleles in the two subclones with different breakpoints, each of which was distinct from the 5q deletion on Clone B, indicative that parallel evolution is indeed reflected in transcription

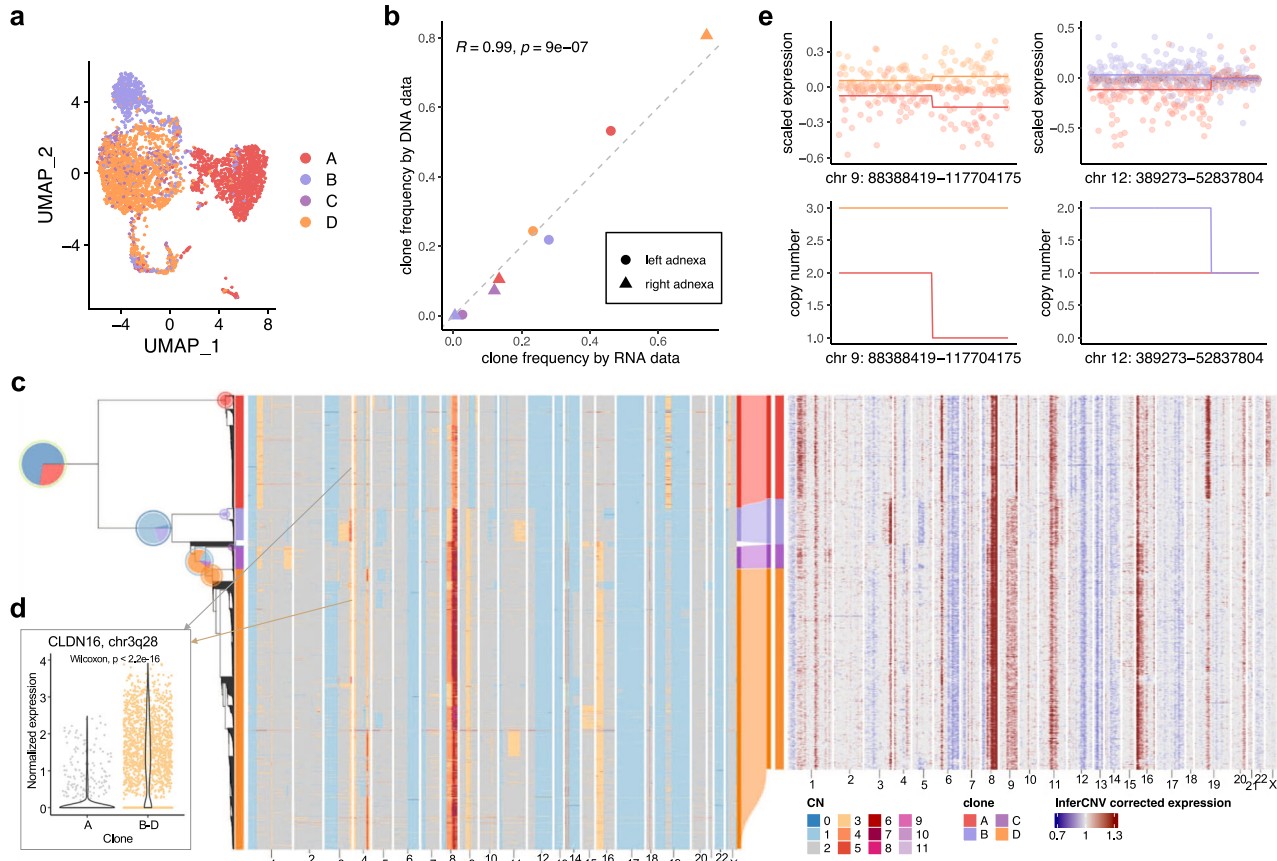

**Fig. 3 | TreeAlign assigns HGSC expression profiles to phylogeny accurately.**
**a** UMAP plot of scRNA-data from patient 022 colored by clone labels assigned by TreeAlign. **b** Correlation between clone frequencies of patient 022 estimated by scRNA-data (x axis) and scDNA data (y axis). Pearson correlation coefficients (*R*) and P values for the linear fit (Two-sided Student's t-test) are shown. **c** Single cell phylogenetic tree of patient 022 constructed with scDNA data (left). Pie charts on the tree showing how TreeAlign assigns cell expression profiles to subtrees recursively. The pie charts are colored by the proportions of cell expression profiles assigned to downstream subtrees. The outer ring color of the pie charts denotes the current subtree. For example, the leftmost pie chart represents the proportions of cells assigned to the two main subtrees. The outer ring represents the root of the phylogeny. The red proportion of the pie chart represents the subtree on the top or clone A. The blue proportion represents the bottom subtree which contains clone B, C and D. Left heat map, total copy number from scDNA; right heat map, InferCNV corrected expression from scRNA; middle Sankey chart, clone assignments from RNA to DNA. **d** Normalized expression of CLDN16 in clone A and clone B-D (Two-sided Wilcoxon signed-rank test). Source data are provided as a Source Data file. **e** Scaled expression and copy number profiles for regions on chromosome 9 and 12 as a function of genes ordered by genomic location.

with the allele specific model (Fig. 4b). Moreover, allele-specific copy number profiles of individual clones (Fig. 4c) were broadly reflected in the allele-specific expression profiles from scRNA data (Fig. 4d). We computed proportions of B allele reads at each heterozygous SNP for each of the subclones assigned from the scRNA data. Subclonal BAF estimated with scDNA data and proportions of reads from B allele from scRNA were significantly correlated ($0.25 < R < 0.53$ for each subclone, $P < 2.2 × 10^{-22}$) (Fig. 4e; Fig. S8c), consistent with more accurate clone assignment. With integrated TreeAlign, we also achieved better performance for predicting allele assignment parameter $a$ of SNPs compared to the allele-specific model (Fig. 4f). We note that recent identifications of parallel allelic-specific alterations whereby maternal and paternal alleles are independently lost or gained in different cells[3,26,33] would further complicate clonal mapping, if allele specificity is not taken into account. Here we show that mono-alleleic expression of maternal and paternal alleles is consistent with coincident maternal and paternal allelic loss in different clones (Fig. 4b). The allele-specific TreeAlign model correctly assigns cells at this level of granularity that would otherwise be missed.

The predicted allele assignments of SNPs from the allele-specific model were consistent with haplotype phasing from scDNA reported by SIGNALS ($AUC = 0.84$). With the integrated model considering total CN expression, the prediction of allele assignments of SNPs can be further improved ($AUC = 0.88$) (Fig. 4f, Methods). We compared the performance of total CN, allele-specific and integrated TreeAlign using subsampled datasets of patient 022 and evaluating against results from the full dataset. Compared to the total CN model, the integrated model performed significantly better when fewer genomic regions were included in the input suggesting it is more robust when there are fewer copy number differences between subclones (Fig. 4g). Both the total CN and integrated model were robust to reduced numbers of cells (Fig. 4h). The allele-specific model without total CN is inferior, as expected.

To investigate the influence of inaccurate phylogeny input on TreeAlign, we randomly selected different proportions of CN profiles from scDNA and shuffled their cell labels in patient 022. With more cell labels being shuffled, the tree will become less accurate in reflecting the true phylogeny of the population. When less than 20% of cells were shuffled, TreeAlign was able to resolve the same number of subclones as with the original data (Fig. S9). When more than 50% cells were shuffled, TreeAlign failed and assigned all expression profiles to the unassigned state. These results suggest that TreeAlign can tolerate inaccurate phylogeny input to some extent.

**Inferring copy number dosage effects in human cancer data**
We next compared the integrated model to the total CN model on a recently published cohort of cell lines and patient derived

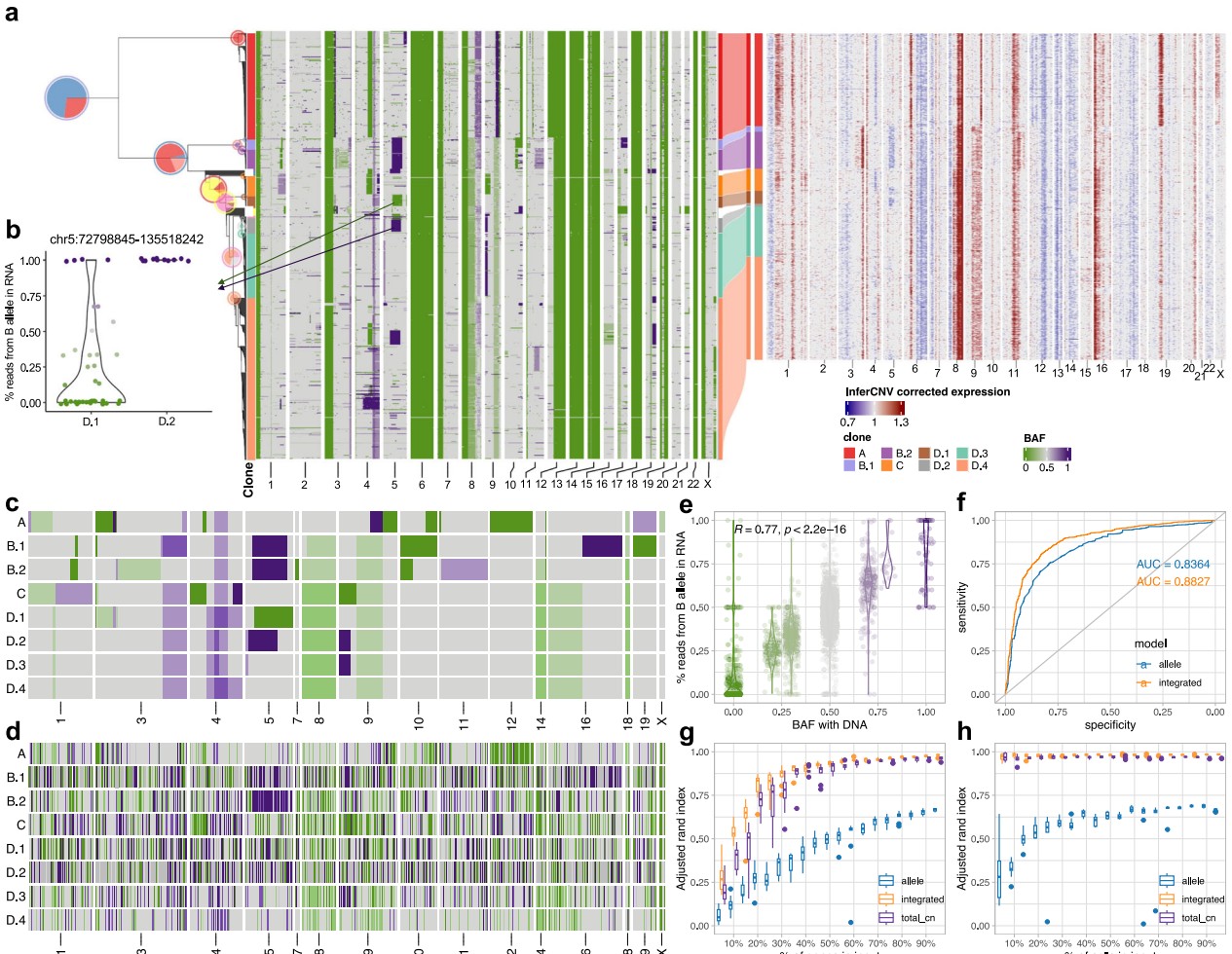

**Fig. 4 | Incorporating allele specific expression increases clone assignment resolution. a** Integrated TreeAlign model assigns expression profiles to phylogeny of patient 022. Left heat map, single cell BAF profiles estimated from scDNA data using SIGNALS, annotated with clone labels on the left side (BAF profiles without clone label represent cells ignored by TreeAlign) (Methods). **b** Proportions of reads from B allele in scRNA-data for clone D.1 and D.2 at region chr5:72,798,845-135,518,242. **c** BAF of subclones with scDNA. Heatmap color represents BAF following the scale in panel **a**. **d** Proportions of reads from B allele for subclones in scRNA. Heatmap color represents the proportion of reads from B allele following the scale in panel **a**. **e**, Correlation between % of reads from B allele in scRNA and BAF estimated with scDNA in patient 022. Annotations at the top indicate the Pearson correlation coefficient ($R$) and $P$ value derived from a linear regression.

Color represents BAF following the scale in panel **a**. **f** ROC curves for predicting $p(a=1)$ with allele-specific TreeAlign and integrated TreeAlign. Haplotype phasing from SIGNALS was treated as ground truth. **g** Robustness of clone assignment to gene subsampling in patient 022. Adjusted Rand index was calculated by comparing clone assignments using subsampled datasets ($n=10$ subsampled datasets for each condition) to the complete dataset. **h** Robustness of clone assignment to cell subsampling in patient 022 ($n=10$ subsampled datasets for each condition). For the box plots in **g**, **h**, box limits extend from the 25th to 75th percentile, while the middle line represents the median. Whiskers extend to the largest value no further than 1.5 times the inter-quartile range (IQR) from each box hinge. Points beyond the whiskers are outliers. Source data are provided as a Source Data file.

xenografts (PDXs) with scDNA and scRNA matched data (Fig. 5a) from Funnell et al.[3]. We applied TreeAlign on data from PDXs of Triple Negative Breast Cancer (TNBC) ($n=3$) and HGSC ($n=6$). In addition, we tested the model on one ovarian cancer control cell line and 6 184-hTERT cell lines engineered to induce genomic instability from a diploid background with CRISPR loss of function of *TP53* combined with *BRCA1* or *BRCA2*. Both integrated and total CN TreeAlign were run on matched DLP+ and 10x scRNA-seq data (Figs. S24–S53). The integrated model was fitted for 1-10 rounds and the total CN model was fitted for 1-3 rounds when we ran TreeAlign with the phylogeny input (Figs. S10 and S11). The integrated model only failed to assign expression profiles to any subclones for cell line SA906a, due to a low number of genes ($n=32$) with CN differences and heterozygous SNPs ($n=7$) with BAF differences between subclones. In comparison, the total CN model failed in 8 cases due to lack of allelic information. As expected, the integrated model characterized more clones

(Fig. 5b) and achieved a lower number of cells not confidently assigned to a subclone (Fig. 5c). For cells that were assigned confidently by the integrated model but not the total CN model, their InferCNV-corrected expression showed higher correlation coefficients with the CN profiles of subclones assigned by the integrated model compared to random subclones (Fig. 5d; Fig. S14), implying better performance of the integrated model.

For high expression genes (mean normalized expression >0.1) located in clone specific copy number (CSCN) regions, 76.7% (64.4–86.6% across cases) had $p(k)>0.5$ suggesting their expression is dependent on copy number (Fig. S15, Fig. S16a). Taking together the simulation results and the fact that there are 13.4–35.6% genes with low CN dosage effects ($p(k)\leq0.5$), we would expect benefits of incorporating $k$ and $p(k)$ in TreeAlign as compared to CloneAlign (Fig. S16c). It was reported that cancer genes tend to have stronger CN-expression correlation compared to non-cancer genes in HGSCs[34]. We also observed concordant results that cancer genes annotated by COSMIC

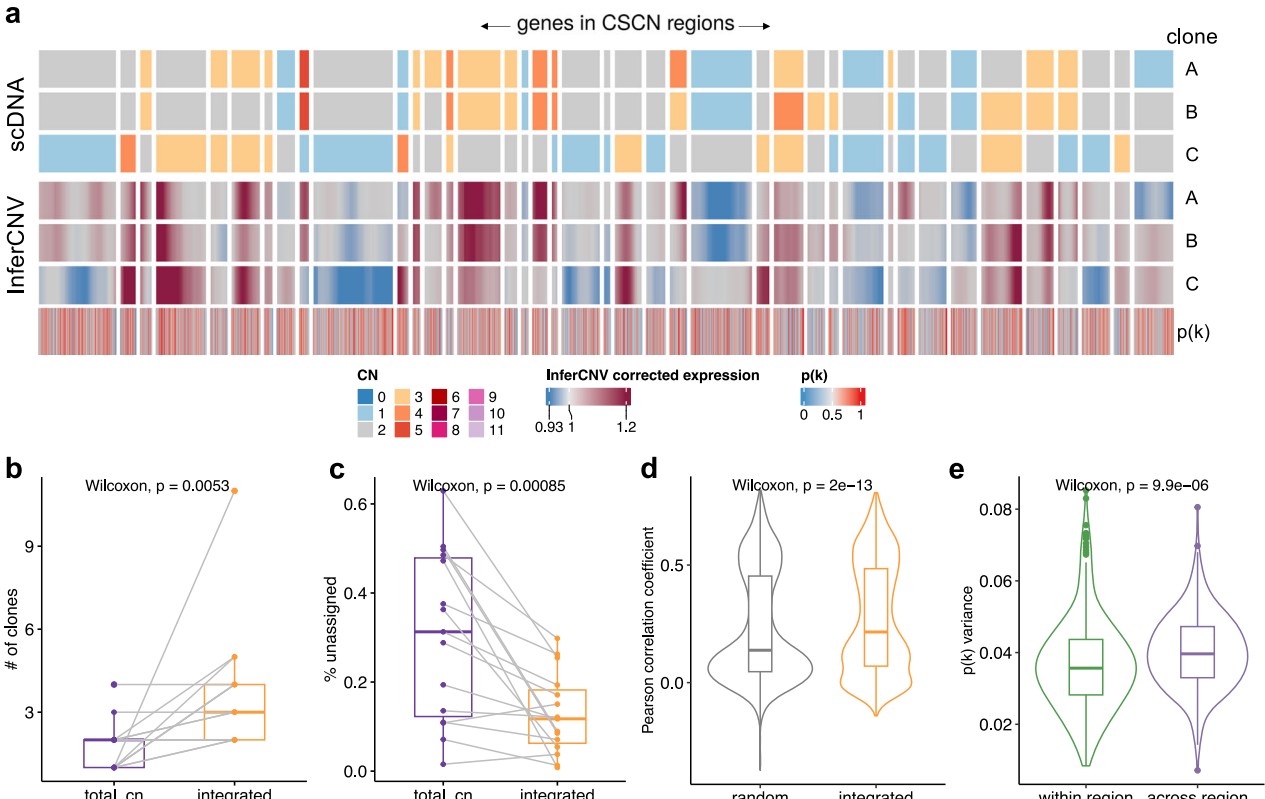

**Fig. 5 | Incorporating allele specific expression increases clone assignment resolution. a** Heat map representations of genes in CSCN regions in HGSC PDX SA1053BX1XB01603. Top heat map: clone-level total CN from scDNA; bottom heat map: InferCNV-corrected expression profiles from scRNA; bottom track: $p(k)$ estimated by TreeAlign. **b** Number of clones characterized by total CN and integrated model ($n = 15$, Two-sided Wilcoxon signed-rank test). **c** Frequencies of unassigned cells (**Methods**) from total CN and integrated model ($n = 15$, Two-sided Wilcoxon signed-rank test). **d** Distribution of Pearson correlation coefficients (R) between scDNA-estimated total CN and InferCNV-corrected expression for cells assigned by the integrated model but unassigned by the total CN model ($n = 8829$ cells, Two-

sided Wilcoxon signed-rank test). Left, correlation distribution calculated by comparing InferCNV profiles to CN profiles of a random subclone; Right, correlation distribution calculated by comparing InferCNV profiles to CN profiles of subclones assigned by integrated TreeAlign. **e** Variance of $p(k)$ sampled from the same genomic regions and across regions ($n = 314$ regions, Two-sided Wilcoxon signed-rank test). For the box plots in **b**–**e**, box limits extend from the 25th to 75th percentile, while the middle line represents the median. Whiskers extend to the largest value no further than 1.5 times the inter-quartile range (IQR) from each box hinge. Points beyond the whiskers are outliers. Source data are provided as a Source Data file.

Cancer Gene Census[35] tend to have higher $p(k)$ compared to non-cancer genes suggesting stronger CN dosage effects in cancer genes (Fig. S16d, e).

When we summarized $p(k)$ by genomic locations, genes located at the same CSCN region had more consistent $p(k)$. Notably, $p(k)$ of genes in a contiguous region exhibited significantly lower variation compared to randomly sampled genes across different regions (Fig. 5a, e). It should be noted that we only included CN events that span more than 10 genes in this analysis. In addition to broad regions of the genome, we note subclonal high-level amplifications affecting known oncogenes which have been identified previously[3]. Using TreeAlign, we also identified subclonal amplifications of oncogenes accompanied by consistent changes in gene expression. For example, in OV2295, subclonal upregulation of MYC expression coincides with the clone-specific MYC amplification with $p(k) > 0.8$ (Fig. S17). To investigate whether MYC pathway activation was also impacted by non-CNA driven effects, we performed pathway enrichment on genes with low $p(k)$ and found genes in the Hallmark MYC Target V1 gene set[36] significantly enriched in low $p(k)$ genes. Combined with HLAMP results, this suggests the pathway can be regulated by both CN dosage effects and other (potentially non-genomic) effects at the subclonal level (Fig. S18a, b), further highlighting the importance of $p(k)$ for interpreting the mechanism of gene dysregulation.

## Clone-specific transcriptional profiles highlight clonal divergence in immune pathways

We next sought to interpret clone-specific transcriptional phenotypes and phenotypic divergence during clonal evolution from TreeAlign mappings. For patient 022, differential expression and gene set enrichment analysis identified genes and pathways upregulated in each clone (Fig. 6a, b). In total, we found 1346 genes significantly upregulated (adjusted $P < 0.05$, MAST[37]) in at least one of the subclones in patient 022. 52.1% (701) of these genes were not located in CSCN regions, while 47.9% (645) genes were located within CSCN regions. For 90.7% (585/645) of genes in CSCN regions, $p(k)$ was > 0.5, reflecting probable gene dosage effects.

Immune related pathways such as IFN-$\alpha$ and IFN-$\gamma$ response were differentially expressed, and with increased relative expression in clone A (Figs. 6c, S19e, S21). Clone A contains cells from both right and left adnexa, thus dysregulation of these pathways cannot be simply explained by the microenvironment of clone A. Differential expression of immune related pathways was also found between more closely related subclones. Compared to clone B.2, clone B.1 also has enriched expression in IFN-$\alpha$ and IFN-$\gamma$ signaling pathways and downregulation in MYC targets V1 and G2M checkpoint gene sets (Figs. S19a, 19f and 22). Clone D.4, compared to other clone D subclones, had downregulated TNF-$\alpha$ signaling via NF$\kappa$B (Figs. S19b, g and S23). Seeking to explain the relative contribution of subclonal CNAs to differentially expressed pathways, we analyzed the proportion of differentially

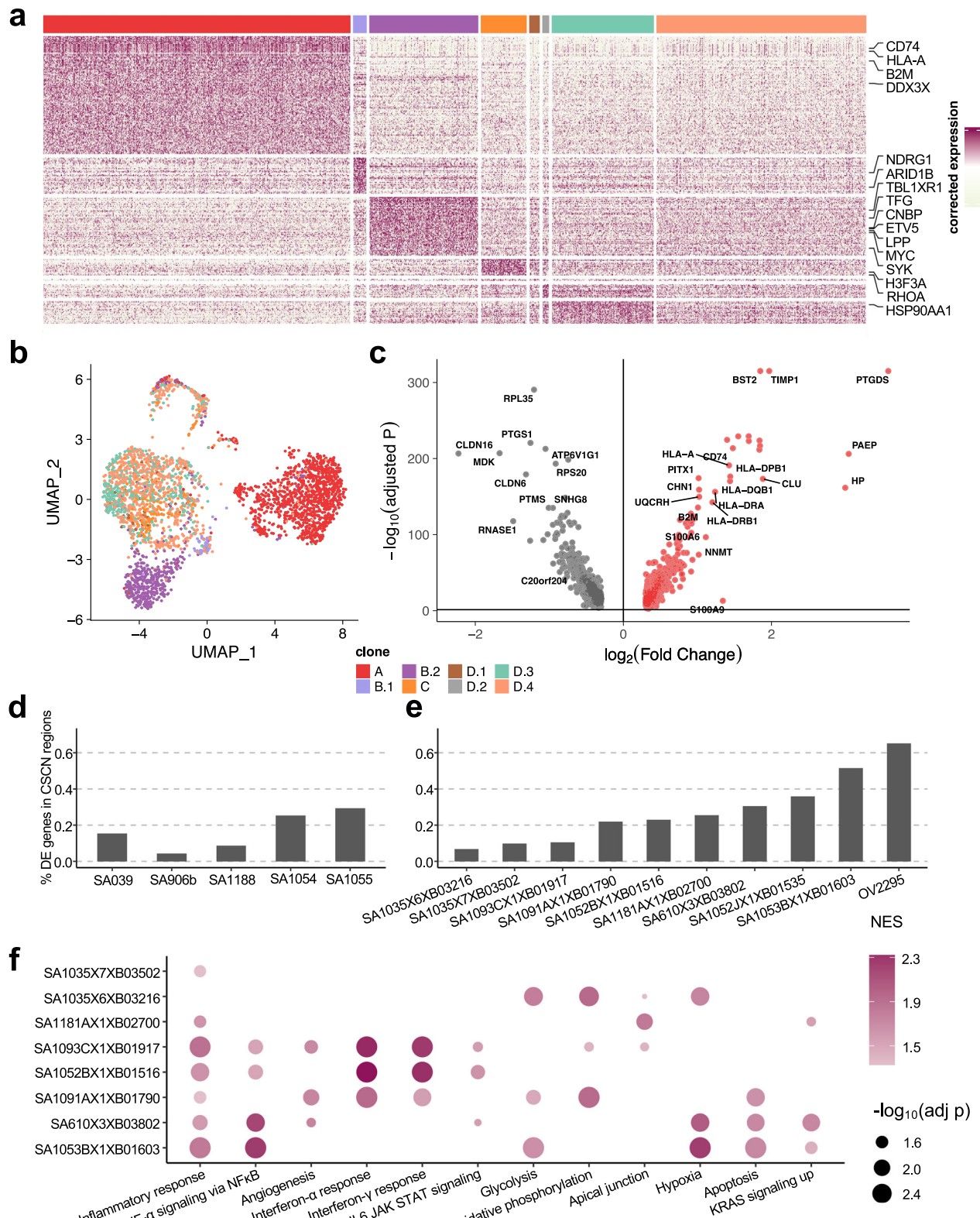

**Fig. 6 | Clone-specific transcriptional profiles highlight clonal divergence in immune pathways. a** Scaled expression of upregulated genes in each subclone in patient 022, showing genes in rows and subclones in columns. Genes in the COS-MIC Cancer Gene Census[35] are highlighted. **b** UMAP embedding of expression profiles from patient 022 colored by clone labels assigned by integrated TreeAlign model. **c** Differentially expressed genes between clone A and other subclones (clone B-D) in patient 022. Adjusted P values were calculated by MAST[37]. Proportions of subclonal differentially expressed genes located in CSCN regions for **d** 184-hTERT cell lines, **e** an HGSC control cell line and PDXs. **f** Pathways with clone-specific expression patterns in TNBC and HGSC PDXs. Source data are provided as a Source Data file.

expressed genes found in subclonal CNAs for each pathway. Only 17.4% (4/23) of differentially expressed genes in the Allograft Rejection gene set are in CSCN regions compared to 61.5% (24/39) in the MYC Targets V1 gene set highlighting the distinct impact of subclonal CNA between pathways (Fig. S20).

We conducted a similar analysis on data from Funnell et al.[3]. Differential expression analysis revealed varying proportions of DE genes located in CSCN regions ranging from 1.3% to 63.9%, indicating that transcriptional heterogeneity due to cis-acting subclonal CNAs varied across tumors (Fig. 6d, e). In addition to pathways such as KRAS signaling, IFN-$\alpha$ and IFN-$\gamma$ response pathways also show frequent variable expression within subclones of TNBC and HGSC PDXs (Fig. 6f). IFN signaling has important immune modulatory effects, and has been previously linked to immune evasion and resistance to immunotherapy[38]. The recurrent differential expression of immune related pathways between subclones suggests their importance in clonal divergence in these cancers of genomic instability.

To investigate transcriptional diversity within and across subclonal populations, we calculated Pearson correlation coefficients ($R$) and Euclidean distance between cells using the top 20 principal components of the gene expression matrices. In addition to TreeAlign, we also used InferCNV to assign cells from scRNA to genomic clones. Clonal frequencies in scRNA estimated by TreeAlign are more consistent with scDNA compared to InferCNV estimations (Fig. S12). We found that cells sampled from the same TreeAlign clone or InferCNV clone tend to have higher correlation and lower distance (Fig. S13), suggesting lower transcriptional diversity within the subclonal populations.

## Discussion

TreeAlign establishes a probabilistic framework for integration of scRNA and scDNA data and inference of dosage effects of subclonal CNAs. TreeAlign achieves high accuracy of assigning single cell expression profiles to genetic subclones and was built to operate on phylogenetic trees directly, therefore informing phenotypically divergent subclones during the recursive clone assignment process. In addition to scRNA and scDNA integration, TreeAlign disentangles the in cis dosage effects of subclonal CNAs which highlights highly regulated pathways in clonal evolution. The model also has improved flexibility allowing either total or allelic copy number or both to be used as input. With additional allele-specific information, TreeAlign has improved prediction accuracy and model robustness and is able to identify more refined clonal structure.

In terms of limitations, TreeAlign was designed to integrate matched scRNA and scDNA datasets. For partially matched datasets with different clonal compositions, TreeAlign may have compromised performance. TreeAlign also assigns expression profiles based on clone-specific CNAs. For cancer types not driven by CN events, TreeAlign is not suitable due to lack of input features. The way TreeAlign encodes the relationship between gene expression and CN could also be further improved. As TreeAlign uses the binary parameter $k$ to indicate whether gene expression is conditioned on copy number, it would be straightforward to change how expected expression is modeled in both conditions and explore more complex patterns of transcriptional regulations in the future. By default, TreeAlign truncates CNs > 10 to 10 and represents the CN-expression relationship with a linear function. Functions that are more biologically meaningful could be used as a replacement. Another extension could be to model $k$ as a categorical variable with a 1 of $K$ Multinomial distribution where the different components represent different functional forms of gene dosage (e.g. linear vs logistic vs exponential).

We expect potential extensions of TreeAlign for integration of other single cell data modalities such as single-cell epigenetic data. Current methods for integration of scRNA and scATAC data are primarily based on nearest neighbor graphs or other distance metrics to match similar cells across multimodal datasets[39]. The advantage of TreeAlign is that it estimates how well the expression of a gene matches with the given biological assumption, hence it is more interpretable and provides explanations for gene expression variations.

The emergence of more single cell multimodal datasets enables future studies to further reveal how genotypes translate to phenotypes and how ongoing mutational processes drive clonal diversification and evolution in cancer cells. It remains an open question whether the CN-expression relation is consistent across tumors and whether application at scale can reveal phenotypic consequences of CNAs at subclonal resolution. Furthermore, as TreeAlign also integrates allele-specific CN and expression, it would be interesting to investigate patterns of LOH and allele-specific expression on a subclone level as modulators of germline alterations and bi-allelic inactivation to better understand these events in the context of tumor heterogeneity and clonal evolution. We expect that concepts introduced in TreeAlign will facilitate the integration of single cell multimodal datasets and the interpretation of associations between modalities.

In conclusion, we anticipate that studying how CNAs impact gene expression programs in cancer applies broadly to different questions in cancer biology including etiology, tumor evolution, drug resistance and metastasis. In these settings, TreeAlign provides a flexible and scalable method for explaining gene expression with subclonal CNAs as a quantitative framework to arrive at mechanistic hypotheses from multimodal single cell data. Our approach provides a computational tool to disentangle the relative contribution of fixed genomic alterations and other dynamic processes on gene expression programs in cancer.

## Methods
### TreeAlign total CN model
The TreeAlign model is a probabilistic graphical model as shown in Fig. 1c. Here we describe the model in detail. Let $X$ be a cell × gene expression matrix of raw counts from scRNA-seq for $N$ cells and $G$ genes, and $x_{ng}$ be the scRNA read count for cell $n$ and gene $g$. Let $\Lambda$ be a gene × clone copy number matrix for $G$ genes and $C$ clones, and $\lambda_{gc}$ be the copy number at gene $g$ for clone $c$. To assign cells from the expression matrix to a clone in copy number matrix, we use a categorical variable $z_n$ which indicates the clone to which a cell should be assigned. $z_n = c$ if cell $n$ is assigned to clone $c$. $z_n$ is drawn from a Categorical distribution with Dirichlet prior.

$$z_{n=1...N} \sim Categorical(\boldsymbol{\pi}) \tag{1}$$

$$\boldsymbol{\pi} \sim Dir(\boldsymbol{\alpha}) \tag{2}$$

To indicate whether the expression of a gene is dependent on the underlying CN, we introduced another indicator variable $k_g$. $k_g = 0$ if expression of gene $g$ is not dependent on CN. $k_g = 1$ if expression of gene $g$ is dependent on CN. $k_g$ is a Bernoulli random variable with Beta prior.

$$k_{g=1...G} \sim Bernoulli(p(k_g)) \tag{3}$$

$$p(k_g) \sim Beta(\beta_1, \beta_2) \tag{4}$$

where we have $\beta_1 = 1$, $\beta_2 = 1$ as default.

Our assumption is that $y_{ng}$, the expected expression of gene $g$ in cell $n$, will be proportional to the copy number of gene $g$ in clone $c$ to which cell $n$ is assigned, if expression of gene $g$ is dependent on copy

number as indicated by $k_g$. Based on this assumption, our model is:

$$y_{ng} = E[x_{ng}|z_n = c] = l_n \times \frac{[\mu_{g0} \times \lambda_{gc} \times k_g + \mu_{g1} \times (1 - k_g)] \times e^{\psi_n \cdot w_g^T}}{\sum_{g'=1}^{G}[\mu_{g'0} \times \lambda_{g'c} \times k_{g'} + \mu_{g'1} \times (1 - k_{g'})] \times e^{\psi_n \cdot w_{g'}^T}} \quad (5)$$

$$X_n = (x_{n1}, \dots, x_{nG}) \quad (6)$$

$$Y_n = (y_{n1}, \dots, y_{nG}) \quad (7)$$

$$X_{n=1\dots N} \sim Multinomial(l_n, Y_n) \quad (8)$$

where $l_n$ is the total scRNA read count from cell $n$. Vector $Y_n$ represents the expected read count for each gene in cell $n$. $X_n$ is the actual read count from each gene in cell $n$ we want to model. $\mu_{g0}$ is the per-copy expression of gene $g$ if the expression is dependent on copy number while $\mu_{g1}$ is the base expression of gene $g$ if its expression is independent of copy number. The intuition is when $k_g = 1$, we expect the expression of $g$ to be proportional to its copy number; when $k_g = 0$, the expression of $g$ is not dependent on the underlying copy number. We specified a softplus transformed Normal prior over the per-copy expression $\mu_{g0}$ and CN-independent base expression $\mu_{g1}$.

$$\mu_{g0}, \mu_{g1} \sim \log(1 + e^{\mathcal{N}(\mu_g', 10)}) \quad (9)$$

where we set $\mu_g'$ to the softplus inverse transformed mean read count of gene $g$ across all cells.

The inner product $\psi_n \cdot w_g^T$ introduces noise into the model to avoid over-fitting. We set the following priors: $\psi_n \sim \mathcal{N}(0, 1)$, $w_{gi} \sim \mathcal{N}(0, \chi_i^{-1})$, $\chi_i$ -$Gamma(2, 1)$[19].

## TreeAlign allele-specific model
To use allele specific copy number information for clone assignment, we set up a separate model, allele-specific TreeAlign which only takes in allele specific information. The input to allele-specific TreeAlign includes single cell level B allele frequencies at heterozygous SNPs estimated from scDNA data and read counts of reference allele and alternative allele of these SNPs from scRNA-data.

Let $t_{ns}$ be the scRNA read count at a heterozygous SNP $s$ in cell $n$, $r_{ns}$ be the scRNA read count from the reference allele at heterozygous SNP $s$ in cell $n$. Both $t_{ns}$ and $r_{ns}$ can be obtained by genotyping heterozygous SNPs of interest with tools such as cellsnp-lite[40].

With scDNA data, we can estimate $b_{sc}$, the BAF for SNP $s$ and clone $c$ using tools such as SIGNALS[3]. We assume that $f_{ns}(z_n = c)$, the expressed reference allele frequency at cell $n$ and SNP $s$ when cell $n$ is assigned to clone $c$ is controlled by the followings: (1). DNA BAF at that SNP of clone $c$ and (2). whether the reference allele in scRNA data is B allele or not. We use a binary variable $a_s$ to indicate whether the reference allele at SNP $s$ should be assigned as B allele. $a_s$ can be obtained using SIGNALS which can use information from scDNA to phase the SNPs in scRNA and assign alleles accordingly. We can also treat $a_s$ as a hidden variable and jointly infer it from the allele-specific model of TreeAlign. Comparing $a_s$ inferred from TreeAlign to SIGNALS output allows us to estimate the performance of TreeAlign.

$$p(a_s) \sim Beta(\beta_1', \beta_2') \quad (10)$$

$$a_{s=1\dots S} \sim Bernoulli(p(a_s)) \quad (11)$$

$$f_{ns}(z_n = c) = a_s \times b_{sc} + (1 - a_s) \times (1 - b_{sc}) \quad (12)$$

$$r_{ns} \sim Binomial(t_{ns}, f_{ns}) \quad (13)$$

where we have $\beta_1' = 1$, $\beta_2' = 1$ as default.

The total CN model and allele-specific model share categorical variable $z_n$ which indicates the clone assignment of cell $n$. Therefore, $z_n$ can be inferred from the two models separately or combined depending on the input data provided. The integrated model is illustrated in Fig. 1c. The prior distributions of all random variables are summarised in (Fig. S1).

## Model implementation and inference
TreeAlign is implemented with Pyro[27] which is a universal probabilistic programming language written in Python and supported by PyTorch. Inference of TreeAlign is done by Pyro's Stochastic Variational Inference (SVI) functions automatically. Specifically, we use the *AutoDelta* function which implements the delta method variational inference[41]. The delta method variational inferences use a Taylor approximation around the maximum a posterior (MAP) to approximate the posterior. Optimization is performed using the Adam optimizer. By default, we set a learning rate of 0.1 and the convergence is determined when the relative change in ELBO is lower than $10^{-5}$ by default.

## Incorporating phylogeny as input
In addition to the gene × clone copy number matrix, TreeAlign can also take the cell × gene copy number matrix from scDNA directly along with the phylogenetic tree constructed from this matrix as input. Starting from the root of the phylogeny, TreeAlign summarizes the copy number of gene $g$ for each clade by taking the mode of copy number, and assigns cells from scRNA to clade-level CN profiles. This process is repeated recursively from the root of the phylogeny to smaller clades until: i) TreeAlign can no longer assign cells consistently in multiple runs (less than 70% cells have consistent assignments between runs by default), or ii) the number of genes located in CSCN regions becomes too small (100 genes in CSCN regions by default), or iii) Limited number of cells remain in scDNA or scRNA (100 by default). By default, TreeAlign also ignores subclades with less than 20 cells in scDNA. Some scRNA cells may remain unassigned to the scDNA phylogenetic tree. For a single cell, if the clone assignment probability < 0.8 or clone assignments are not consistent in 70% of repeated runs, the cell will be denoted as unassigned. This feature is important to the model because there might be incomplete sampling of a given tumor, leading to a subclone only appearing in one of the two data modalities. Note, all parameters are fully configurable at run time by the user.

## Benchmarking clone assignment and dosage effect prediction with simulations
To generate simulated data for benchmarking, TreeAlign model was fit to the MSKSPECTRUM patient 081 dataset to obtain the empirical estimations of model parameters. Then we simulated from TreeAlign considering the following scenarios: 1. Varying proportion (10%, 20%, 30%, …, 90%) of genes with dosage effect. 2. Varying number of genes (100, 500 and 1000) in CSCN regions. 3. Varying number of cells (100, 1000 and 5000) in scRNA.

We compared TreeAlign to CloneAlign and InferCNV v.1.3.5 in terms of the performance of clone assignment. For CloneAlign, we summarized clone-level copy number by calculating the mode of copy number for each gene and ran CloneAlign with default parameters. For InferCNV, we used the recommended setting for 10x. 3200 non-cancer cells were randomly sampled from the MSK SPECTRUM dataset and used as the set of reference "normal" cells. To assign clones with InferCNV, we calculated Pearson correlation coefficient between InferCNV corrected gene expression profile (expr.infercnv.dat) and the clone-level copy number profiles from scDNA. Cells from scRNA-seq were assigned to the clone according to the highest correlation coefficient. Accuracy of clone assignment was computed to compare the performance of the three methods. We also evaluated TreeAlign's performance on predicting CN dosage effects.

To evaluate TreeAlign's performance on predicting CN dosage effects, we calculated the area under the curve (AUC) using $p(k)$ output by TreeAlign, and compared it to a baseline model. The baseline model for CN dosage effects was constructed by (1). assigning expression profiles to genomic clones using CloneAlign (2). calculating Pearson correlation coefficients ($R$) between normalized read count from scRNA and clone-specific CN from scDNA for each gene in input. The resulting $R$ can be viewed as a metric for CN dosage effects. We calculated the baseline model AUC using $R$ and compared to TreeAlign model.

To demonstrate the performance of allele-specific TreeAlign, for the simulated datasets with 30% CN-dependent genes, we also simulated reference allele and total read counts for varying number of heterozygous SNPs (0, 250, 500, 750, 1000 and 1250) from the generative model of allele-specific TreeAlign. Adjusted Rand index of clone assignments was calculated to evaluate the performance of the integrated TreeAlign model on simulated datasets with varying number of heterozygous SNPs.

To evaluate TreeAlign's performance on inaccurate trees, we randomly shuffled labels for different proportions (10%, 20%,...,90%) of cells on the phylogeny of patient 22. TreeAlign was run with the shuffled phylogenies. Clone assignment results were compared to results obtained from the original phylogeny using adjusted Rand index.

## Human participants

All patient data were obtained from the MSK SPECTRUM cohort published before. Information regarding patient consent and ethical approval can be found in the previous publication[7].

## MSK SPECTRUM data

We obtained matched scRNA and scDNA from two HGSC patients (patient 022 and patient 081) from the MSK SPECTRUM cohort[7]. Samples were collected under Memorial Sloan Kettering Cancer Center's institutional IRB protocol 15-200 and 06-107. Single cell suspensions from surgically excised tissues were generated and flow sorted on CD45 to separate the immune component. CD45 negative fractions were then sequenced using the DLP+ platform[3,6,30].

## Gastric cancer cell line data

Preprocessed scDNA data and scRNA count matrix of the gastric cancer cell line (NCI-N87)[32] were downloaded from SRA (PRJNA498809) and GEO (GSE142750). Copy number calling for scDNA was performed using the Cellranger-DNA pipeline using default parameters.

## PDXs and additional cell line data

scRNA and scDNA from 6 HGSC PDX samples (SA1052BX1XB01516, SA1052JX1XB01535, SA1053BX1XB01603, SA1091AX1XB01790, SA1093CX1XB01917, SA1181AX1XB02700), 3 TNBC PDX samples (SA1035X6XB03216, SA1035X7XB03502, SA610X3XB03802), 1 ovarian cancer cell line (OV2295) and 6 hTERT-184 cell lines (SA039, SA1054, SA1055, SA1188, SA906a, SA906b) were downloaded from https://zenodo.org/records/6998936 according to Funnell et al.[3].

## scDNA data analysis

scDNA DLP+ data was processed[3,30] using the Isabl platform[42]. Single cell copy number was inferred using HMMcopy[43]. Cells with quality score > 0.75 and not in S-phase were retained for downstream analysis. Allele specific copy number was called using SIGNALS, which provides allele specific copy number of the from A|B in 500kb bins across the genome. A and B being the copy number of alleles A and B respectively with *total CN = A + B*. As the single cell data is sparse, only a subset of germline SNPs have coverage in each cell, therefore to produce the input required for TreeAlign (B-Allele frequencies per SNP per cell), we impute the BAF of each SNP assuming that a SNP will have the same BAF as the bin in which the SNP resides.

## Clustering and phylogenetic inference

Clustering and phylogenetic inference of scDNA was performed using UMAP and HDBSCAN (parameters min_samples=20, min_cluster_size=30, cluster_selection_epsilon=0.2). For patient 022, we also constructed phylogenetic trees using Sitka[6].

## Genotyping SNPs in scRNAseq cells

SNPs identified in scDNA-seq and matched bulk whole genome sequencing were genotyped in each single cell using cell-snplite with default parameters.

## scRNA data analysis

For scRNA data processing, read alignment and barcode filtering were performed by CellRanger v.3.1.0. Cancer cell identification was performed with CellAssign[44]. Principal-component analysis (PCA) was performed on the top 2000 highly variable features output by function FindVariableFeatures using Seurat v.4.2[45]. UMAP embeddings and visualization were generated using the first 20 principal components. Unsupervised clustering was performed using FindNeighbors function followed by FindClusters function (resolution = 0.2). To compare transcriptional heterogeneity across or within clones, we randomly sampled 100 expression profiles from the following groups: 1. all cancer cells in a patient/cell line/PDX 2. cancer cells in the same TreeAlign clone 3. cancer cells in the same InferCNV clone. Pearson correlation coefficients and Euclidean distance between the sampled expression profiles were calculated using the top 20 principal components.

## Differential expression and gene set enrichment analysis

Differential expression analysis was performed using FindAllMarkers and FindMarkers function (test.use="MAST", latent.vars=c("nCount_RNA", "nFeature_RNA")) in Seurat v.4.2. Only G1 cells were used in differential expression analysis to avoid confounding of cycling cells. Cell cycle phase was annotated with CellCycleScoring function in Seurat.

We used the fgsea v.1.24.0[46] package to conduct gene set enrichment analysis with Hallmark gene sets ($n$=50) downloaded from MSigDB[47]. We set the following parameters for the gene set enrichment analysis: nperm = 1000, minSize = 15, maxSize = 500.

## Statistical analysis and visualization

Statistical tests and visualization were performed with R (v.4.2) package ggpubr (v.0.5.0) and ggplot2 (v.3.4).

## Reporting on sex and gender

Data from patient 081 and patient 022 from study cohort MSK SPECTRUM were used in this study. All patients are female with high-grade serous ovarian cancer.

## Reporting summary

Further information on research design is available in the Nature Portfolio Reporting Summary linked to this article.

# Data availability

Processed data containing input and output of TreeAlign have been deposited in Zenodo [https://doi.org/10.5281/zenodo.7517412]. Raw scDNA data and scRNA count matrix of the gastric cancer cell line (NCI-N87) can be accessed from SRA (PRJNA498809) [https://www.ncbi.nlm.nih.gov/bioproject/?term=PRJNA498809] and GEO (GSE142750) [https://www.ncbi.nlm.nih.gov/geo/query/acc.cgi?acc=GSE142750]. Raw scDNA and scRNA data from Funnell et al. are available at [https://ega-archive.org/studies/EGAS00001006343]. Raw scRNA data for patient 022 and patient 081 are available at GEO (GSE180661) [https://www.ncbi.nlm.nih.gov/geo/query/acc.cgi?acc=GSE180661]. Hallmark gene sets were downloaded from MSigDB [https://www.gsea-msigdb.

org/gsea/msigdb/human/genesets.jsp?collection=H]. Source data are provided as a Source Data file. Source data are provided with this paper.

## Code availability

The code is publicly accessible on a GitHub repository [https://github.com/shahcompbio/TreeAlign], which implements TreeAlign and describes how to generate simulated datasets.

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

## Acknowledgements

This project was funded in part by Cycle for Survival supporting Memorial Sloan Kettering Cancer Center. SPS holds the Nicholls Biondi Chair in Computational Oncology and is a Susan G. Komen Scholar. This work was funded in part by the Cancer Research UK Grand Challenge Program to SPS [C42358/A27460], 1P50CA247749-01 awarded to SPS, a National Institutes of Health Center for Excellence in Genome Sciences grant RM1-HG011014, Break Through Cancer and the NCI Cancer Center Core Grant P30-CA008748.

## Author contributions

S.P.S. conceived the study design. H.S. designed the statistical method. H.S. and A.C.W. implemented the software. H.S., M.J.W. and A.C.W. performed the data processing, analysis, and simulations. H.S., S.P.S., A.C.W., M.J.W., A.M. and G.S. wrote and reviewed the final manuscript.

## Competing interests

SPS is a consultant to AstraZeneca Inc., outside the scope of this work. SPS received funding from Bristol Meyers Squibb Inc. The remaining authors declare no competing interests.
