## [Peer Review File · Nature Communications]

REVIEWER COMMENTS

Reviewer #1 (Remarks to the Author): Expert in clonal evolution inference methods, single-cell multi-omics, and bioinformatics

This manuscript introduces TreeAlign, a computational method to map unpaired scRNA-seq and scDNA-seq data. The method extends the basic Clonealign (Campbell et al, 2019) model by adding a per-gene dosage effect indicator variable. It can also take as input B-allele frequencies estimated separately from the scDNA data, to model allele-specific expression and thus improve the mapping slightly. Finally, the method may take a phylogenetic tree as input in order to avoid fixing the clone labels a priori from the scDNA data. In this case, it fits the model many times to clones defined at different depths in the tree and reports the highest level of granularity at which the model fit was satisfactory.

The method works as expected in simulations but lacks validation on real data, hence it remains unclear whether it provides any relevant advance. In addition, the methodological novelty is limited and the modelling choices are not entirely clear. The manuscript is generally well-written but some exposition issues have to be addressed. We detail our concerns in the following paragraphs.

Comments on the results

The text in this section is well-written and easy to follow. The simulations show that the method outperforms alternatives in the realistic scenario that not all genes have dosage-dependent events. However, the real data results are purely observational and the authors do not seek any validation for the inferred gene dosages from their model. Are the genes detected by their model known to be unaffected by dosage? Is there a known regulatory mechanism that explains the ones which are predicted to be dosage-sensitive versus the ones that are not? In general, the authors simply show that the model outputs results for the new parameters that can be interpreted in some way, but there is no clear evidence that the results are in line with previous knowledge, in particular for the dosage.

In simulations, TreeAlign outperforms Clonealign when the true % of genes with CN dosage effect is less than 90%. The authors should highlight what could be expected from real data based on their model fits so that the benefit from TreeAlign for real data is evident.

The authors do not evaluate the allelic imbalance section of their model in simulations. The authors must include an assessment that shows how the accuracy of cell-to-clone assignments changes when this information is leveraged.

The authors should use a baseline for gene dosage estimation. A simple alternative would be the per-gene correlation of expression and CNVs after fitting Clonealign.

The authors apply TreeAlign to an HGSC patient. They first assign cells to 4 clones. The InferCNV profiles correlated well with the 4 assigned CNV clones. They conclude that using scDNA data for mapping is better than just using InferCNV because it misses important events. This is not surprising, but still, a

welcome confirmation that performing scDNA-seq is worth it in studies of the genotype-phenotype relationship. They check the assignment accuracy by comparing clonal frequencies, as in the Clonealign and SCATrEx papers.

Then, they check whether using allele-specific expression helps the assignments. First, they use only allele-specific expression, and then they integrate with total CN too. The allele-specific-only model is highly concordant with the total CN model. The integrated model was able to go deeper into the tree for the assignments, which shows the benefit of their iterative model fitting framework.

They then assessed the ability to infer dosage effects in real data. They observed that dosage effects had a spatial structure in the genome, which might just be a technological artifact: the CNVs detected with the DLP+ protocol will typically span many genes and no focal events. This should be discussed in the text.

They found a positive relationship between MYC amplification and over-expression (i.e., a high dosage effect). They then performed pathway enrichment of genes with low dosage effect, and found enrichment for genes in the MYC pathway. They argue that this shows that MYC pathway is activated both via copy number changes directly and by non-copy number changes indirectly. The authors should clarify how their method enables these conclusions to be reached more easily than with a simple alternative of performing pathway enrichment in genes for which the copy number is similar across clones, as in the Clonealign paper.

Finally, they checked clone-specific transcription. They do this by DE and GSEA on each clone, and then check which genes fall in copy-number effect regions and which don't. This is the same procedure as in the Clonealign paper.

Comments on the method and its exposition

The main novelty and benefit of TreeAlign reside mostly in the ability to identify gene dosage effects. Methodologically, this consists of adding one additional binary random variable to the Clonealign model, which is a simple extension. It is unclear whether the most reasonable way to model gene dosage is via a binary variable, instead of a continuous one reflecting the different sensitivities that different genes have to copy numbers.

The allelic imbalance model is a useful addition, even though it does not make a substantial improvement to the cell-to-clone assignments. For the allelic imbalance, do they only care about cases with LOH and CN-neutral LOH, or do they consider other cases with allelic imbalance without LOH? E.g. 3 copies of one allele and 1 copy of another.

The allele-specific model is not described in detail, and only an overview is given. The authors must describe the complete Bayesian model. Additionally, they state their model is a Bayesian one, but do not show the prior distributions for the random variables in their model, or indeed even specify clearly which are random variables and which are learnable parameters. Is there a prior on the per-copy

expression μ_{g0} ? And on the random effects part? How are they optimized? What are Delta distributions? The authors mention variational inference, but they also mention MAP estimation. Which do they use? What is the input to the allele-specific model? Some of this information is scattered between Figure 1, Table 1, and the Methods section, with no explicit correspondence. The authors must provide a complete and detailed description of the method in the Methods section.

In the model (Equation 3), the noise factors are only active for genes with no clonal effects. Perhaps they are missing a parenthesis to make sure all terms are affected?

In the graphical model (Figure 1 c), what is the random variable γ ? Where is its dependence on the parent nodes described?

Line 551: What does it mean for the indicator variable a_g to be “optional”? And how does this variable relate to the assignment model? What are all the other variables in the allele-specific model shown in Figure 1 c and how do they relate to a_g ?

Lines 555-556: The authors state that the total-CN and the allele-specific models were combined for the integrated TreeAlign model, but provide no details on how they were combined formally.

How does the model deal with dosage of very high copy numbers? Presumably, a gene with 10 copies might have a lower than 5-fold increase in gene expression through some mechanism of regulatory compensation, as described in the Clonealign paper. Do the authors also truncate the input CNVs to their model?

The method may take as input a single-cell phylogenetic tree with copy numbers, which may be unreliable at the single-cell level due to the sparse nature of whole-genome scDNA-seq data, and the difficulty of building a tree from it. While this is somewhat bypassed through the recursive fitting scheme employed by the authors, a discussion of the effect of the accuracy of the tree structure used as input should be included.

Since the method involves multiple rounds of fitting a Bayesian model to data with different combinations of subclonal copy number profiles as input, the authors must report how many rounds of fitting they have to go through for each tree in the real data.

Can the iterations of fitting the Bayesian model deeper in the tree be initialized from the results from previous iterations at lower depths in the tree? This might significantly reduce computation time, instead of re-initializing the fit from scratch every time.

The authors state that alternative methods, including Clonealign, SCATrEx and CCNMF “require using predefined subclones from scDNA data as input”. On the one hand, this is not true for CCNMF, which takes as input single-cell copy number profiles, instead of subclones, and finds a clustering that matches both the scRNA and the scDNA data simultaneously. On the other hand, the TreeAlign model as described in the Methods section also takes predefined subclones from scDNA data. The only difference is in the pre-processing and post-processing that selects the appropriate subclones, by using simple

criteria such as the number of cells in a subclone from the scDNA data, or the percentage of scRNA cells confidently assigned to subclones.

The use of the phylogenetic tree to define the depth at which clones should be defined for effective mapping between the two technologies can essentially be seen as a wrapper function around Clonealign that checks the quality of the model fit to stop increasing the subclonal granularity. This procedure can be easily applied to any method that performs the scRNA-scDNA mapping, including Clonealign, SCATrEx and, in principle, CCNMF.

We thus believe that the authors must rephrase their method overview to make this aspect explicit. For example, stating that “TreeAlign is a Bayesian probabilistic model that maps gene expression profiles from scRNA to phylogenies from scDNA which obviates the need to identify clones a priori from a tree” is not accurate. Rather, “TreeAlign is a software package that fits a Bayesian probabilistic model that maps gene expression profiles from scRNA to clones from scDNA that are refined recursively based on the quality of the model fit at each level of the scDNA phylogenetic tree” would be a fairer description. To be clear, we believe this procedure is a good idea, but its scope should be evident in the text.

Comments on the figures

Figure 1: not all variables in the graphical model are described in the text: t , r , b , and π , for example.

Figure 2 b: it is confusing to have the nodes and the clones in the figure. Why not just label the clones? Cells are assigned to clones, right? Plus, in subsequent figures nodes are not labelled anymore, only clones.

Figure 2 c: using percentages to denote expression levels is unclear. Perhaps the authors can make it more explicit what the facets mean in the figure.

Figure 3 d and Figure 4 a: please clarify the meaning of the colors in the pie charts. What does blue encode, if there is no blue clone?

Figure 4 d: Why are the scRNA BAFs so heterogeneous compared to the scDNA ones? How are these estimated?

Figure 4f: How do they compute the AUC? What's the ground truth?

Figure 5 a: Please label the rows and columns of the heatmaps.

Figure 6 a: The caption reads that the rows are genes and the columns are clones, but the rows are also coloured by clones. Why?

Figure 6 b, c, and d: the caption labels do not match the subfigures. Example: the legend says b-c contains proportions of DE genes, but b contains a UMAP and c contains a volcano plot. This also happens in Figure 9.

References

Campbell, K. R. et al. clonealign: statistical integration of independent single-cell RNA and DNA sequencing data from human cancers. *Genome Biol.* 20, 54 (2019).

Reviewer #2 (Remarks to the Author): Expert in cancer single-cell genomics and evolution, ovarian cancer, and immune cell landscapes

The manuscript by Shi, et al solves the unmet need to link subclonal CNAs to clone-specific phenotypes using single cell sequencing. As some cancers are driven in large part by CNA (versus somatic mutations), this is an important area of study. The method can also improve subclonal assignments as well as enabling the inference of genes that are mechanistically altered through subclone specific CNAs, and identification of expression programs that are genomically independent. This tool is really quite clever. One challenge for application of the method is that most datasets, especially patient samples, lack both scRNAseq and scDNAseq data, and more frequently combine bulk DNAseq with scRNAseq, for example. However, perhaps this is a first step towards leveraging alternative data types. I only have some minor to moderate suggestions, mostly aimed at clarifying functionality and details of the method. Overall, this is an important new and original approach. The study could be strengthened by providing a broader summary of results across patients/datasets as a whole (instead of highlighting specific samples) and a more in depth description of the method.

1. For the patient and cell line data, it would be helpful to assess the relationships between the inferCNV (or copykat) CNVs with the TreeAlign method as a whole across all patients/etc (both advantages, which there clearly are, as well as any results the detail performance of the method when applied to different data types/quality).
2. Figure 3D and 4A are beautiful and really highlight the strength of the method. However, is this result typical across patients? It would be helpful if the additional patient results are included as supplemental files/data. When does the approach fail (low read count, minimum number of cells, etc)? What parameters are important for linking data effectively? In general, the methods (and results) could be expanded on and more detailed.
3. In addition, both of these datasets include relatively small numbers of cells--are there larger cohorts that could also be example from additional CNA rich tumors to provide a broader survey of the method's applicability, advantages and weaknesses?
4. The phenotype examples are somewhat peripheral to the study and it is currently hard to assess the accuracy of the TreeAlign approach compared to others. Is there a way to calculate the phenotypic diversity within and across subclonal populations using the TreeAlign versus InferCNV subclonal calls? The hypothesis would be that the cells from each subclone have more similar phenotypes to each other than to cells from other subclones.

Reply to reviewer reports on Nature Communications submission of the manuscript:

“Exploiting allele-specific transcriptional effects of subclonal copy number alterations for genotype-phenotype mapping in cancer cell populations”

Summary

We would like to thank the reviewers for careful consideration of our manuscript NCOMMS-23-02664-T, titled “*Exploiting allele-specific transcriptional effects of subclonal copy number alterations for genotype-phenotype mapping in cancer cell populations*”. We have revised the manuscript to include additional simulations to further validate TreeAlign’s performance, more detailed analyses of TreeAlign results on cancer datasets and comprehensive description of the model in **Methods**. These modifications have further strengthened the work and resulted in a much improved presentation of TreeAlign and the biological insights gained by the method.

In brief, we have revised the manuscript as follows:

- We included additional simulation analyses:
 - We synthesized datasets with heterozygous SNPs to confirm that the allele-specific data are informative to clone assignments.
 - We compared CN dosage effects from TreeAlign to a baseline model and showed that TreeAlign had higher accuracy in predicting CN dosage effects.
 - We simulated inaccurate phylogenies by shuffling cell labels in patient 22 and demonstrated that TreeAlign is robust to tree inaccuracy.
- We updated **Methods** and included new analysis results on cancer datasets:
 - We included more description of the model setup and prior distributions of random variables in **Methods**
 - Additional correlation analysis was included and showed that the CN dosage effect prediction $p(k)$ tends to be higher for cancer genes in concordance with previous studies.
 - We evaluated transcriptional heterogeneity within and across subclones and showed that cancer cells from the same subclone had lower transcriptional diversity.

The additional simulation analyses and deeper analyses of cancer datasets further support our conclusions. Detailed changes to our manuscript and point-by-point responses to the comments of the reviewers are included below.

Referees' comments:

[C=comment, R=reply]

Referee #1

This manuscript introduces TreeAlign, a computational method to map unpaired scRNA-seq and scDNA-seq data. The method extends the basic Clonealign (Campbell et al, 2019) model by adding a per-gene dosage effect indicator variable. It can also take as input B-allele frequencies estimated separately from the scDNA data, to model allele-specific expression and thus improve the mapping slightly. Finally, the method may take a phylogenetic tree as input in order to avoid fixing the clone labels a priori from the scDNA data. In this case, it fits the model many times to clones defined at different depths in the tree and reports the highest level of granularity at which the model fit was satisfactory.

The method works as expected in simulations but lacks validation on real data, hence it remains unclear whether it provides any relevant advance. In addition, the methodological novelty is limited and the modelling choices are not entirely clear. The manuscript is generally well-written but some exposition issues have to be addressed. We detail our concerns in the following paragraphs.

Major comments

[C1]: The text in this section is well-written and easy to follow. The simulations show that the method outperforms alternatives in the realistic scenario that not all genes have dosage-dependent events. However, the real data results are purely observational and the authors do not seek any validation for the inferred gene dosages from their model. Are the genes detected by their model known to be unaffected by dosage? Is there a known regulatory mechanism that explains the ones which are predicted to be dosage-sensitive versus the ones that are not? In general, the authors simply show that the model outputs results for the new parameters that can be interpreted in some way, but there is no clear evidence that the results are in line with previous knowledge, in particular for the dosage.

[R1]: Due to lack of co-registered multimodal single cell datasets, the dosage effects from subclonal copy number change and potential regulatory mechanisms are less well studied. From analysis of bulk sequencing data, it was reported that cancer genes tend to have stronger CN-expression correlation compared to non-cancer genes in HGSCs¹. We also observed concordant results that cancer genes annotated by Cancer Gene Census tend to have higher $p(k)$ compared to non-cancer genes suggesting stronger CN dosage effects in cancer genes (**Extended Data Fig. 15d, e, manuscript Line 265**). While we appreciate the comments on mechanisms, we do not claim mechanistic discovery in this manuscript and suggest this would be out of scope for the current contribution. Indeed, we have additional studies underway to investigate mechanisms, however this experimental data will take some years to complete.

Extended Data Fig. 15. **Distribution of $p(k)$ in PDXs and cell lines.** d-e, $p(k)$ for cancer genes and non-cancer genes in (d) PDXs and cell lines and (e) patient 022

[C2]: In simulations, TreeAlign outperforms Clonealign when the true % of genes with CN dosage effect is less than 90%. The authors should highlight what could be expected from real data based on their model fits so that the benefit from TreeAlign for real data is evident.

[R2]: From TreeAlign results on the PDX and cell line datasets, we observed 10%-35% genes with low dosage effects ($p(k) < 0.5$), suggesting that for most cases, there are less than 90% genes with CN dosage effects (**Extended Data Fig. 15c**). It was also estimated that out of 7641 genes in the GEO and TCGA dataset, 38.7% of genes had high degree transcriptional adaptation or low correlation between CN and expression². With synthetic datasets, TreeAlign showed significantly improved accuracy when there were less than 90% genes with CN-dependent dosage effects. Therefore, the benefit of TreeAlign should be evident for real data.

Extended Data Fig. 15. **Distribution of $p(k)$ in PDXs and cell lines.** c, Proportions of genes with low CN dosage effects ($p(k) < 0.5$) in PDXs and cell lines.

[C3]: The authors do not evaluate the allelic imbalance section of their model in simulations. The authors must include an assessment that shows how the accuracy of cell-to-clone assignments changes when this information is leveraged.

[R3]: To investigate whether allele-specific information is useful for clone assignment, we sampled the B allele frequencies of various numbers (0, 250, 500, 750 and 1000) of heterozygous SNPs from patient 081 scDNA. We simulated scRNA reference allele read count of these heterozygous SNPs from the generative model of TreeAlign. We evaluated TreeAlign's performance on these datasets and found that clone assignment accuracy was improved when more SNPs were included in the simulated datasets (**Extended Data Fig. 5**).

Extended Data Fig. 5. **Clone assignment accuracy with simulated allelic data.** Accuracy of clone assignment for the integrated model of TreeAlign on simulated scRNA datasets as a function of varying numbers of heterozygous SNPs in input. Panels represent datasets with different numbers of genes and proportions of genes with CN dosage effects.

[C4]: The authors should use a baseline for gene dosage estimation. A simple alternative would be the per-gene correlation of expression and CNVs after fitting Clonealign.

[R4]: As suggested, we compared $p(k)$ to a baseline estimation of CN dosage effects which is the per-gene Pearson correlation coefficient (R) of CN and expression after fitting CloneAlign.

p(k) from TreeAlign had an overall higher AUC compared to R from CloneAlign for predicting CN dosage effects (Extended Data Fig. 4).

Extended Data Fig. 4. **Dosage effect prediction of TreeAlign in simulated datasets.** AUC of CN dosage effect p(k) predicted by CloneAlign (baseline model) and TreeAlign as a function of gene expression level. Genes were assigned to 10 bins based on expression level. Ranges of normalized expression for each bin were shown in brackets. Panels represent simulated datasets with varying gene dosage effect frequencies.

[C5]: The authors apply TreeAlign to an HGSC patient. They first assign cells to 4 clones. The InferCNV profiles correlated well with the 4 assigned CNV clones. They conclude that using scDNA data for mapping is better than just using InferCNV because it misses important events. This is not surprising, but still, a welcome confirmation that performing scDNA-seq is worth it in studies of the genotype-phenotype relationship. They check the assignment accuracy by comparing clonal frequencies, as in the Clonealign and SCATrEx papers.

Then, they check whether using allele-specific expressions helps the assignments. First, they use only allele-specific expressions, and then they integrate with total CN too. The allele-specific-only model is highly concordant with the total CN model. The integrated model was able to go deeper into the tree for the assignments, which shows the benefit of their iterative model fitting framework.

They then assessed the ability to infer dosage effects in real data. They observed that dosage effects had a spatial structure in the genome, which might just be a technological artifact: the CNVs detected with the DLP+ protocol will typically span many genes and no focal events. This should be discussed in the text.

[R5]: We agree that focal events are less likely to be captured by DLP+ compared to CNAs that span large regions. In PDX/cell line data, most of the CSCN regions are 500 kb which is also the smallest region to confidently call copy numbers using DLP+ data. There are on average 4.22 genes in these 500 kb CSCN regions with an average $p(k)$ of 0.57 which is lower compared to larger regions with average $p(k) = 0.64$ (**Rebuttal Figure 1**).

To investigate the genomic spatial structure of dosage effects, we only included CSCN regions that span more than 10 genes in the analysis and noticed that $p(k)$ from the same genomic region tend to be more similar and have lower variations. As focal events shorter than 500 kb are more likely to be missed by DLP+, it is not known whether this conclusion still holds for these focal copy number events. We added related discussions on **Line 270** to emphasize that the observation was only made for large genomic regions.

Rebuttal Fig. 1. **CSCN region length in PDXs and cell lines.** a, Length distribution of CSCN regions in PDXs and cell lines. b, $p(k)$ of genes in 500 kb CSCN regions and other longer regions.

[C6]: They found a positive relationship between MYC amplification and over-expression (i.e., a high dosage effect). They then performed pathway enrichment of genes with low dosage effect, and found enrichment for genes in the MYC pathway. They argue that this shows that the MYC pathway is activated both via copy number changes directly and by non-copy number changes indirectly. The authors should clarify how their method enables these conclusions to be reached more easily than with a simple alternative of performing pathway enrichment in genes for which the copy number is similar across clones, as in the Clonealign paper.

[R6]: With TreeAlign output, we found that genes in the MYC pathway tend to have lower $p(k)$ suggesting the MYC pathway was also regulated by mechanisms other than cis regulation of CNAs. It is possible to conduct similar analyses using CloneAlign by estimating correlation between CN and expression post clone assignments. However, we demonstrated through simulations that this approach is less optimal compared to TreeAlign. The analysis of finding enriched pathways from clone-specific differentially expressed genes located outside of CSCN

regions as described in the CloneAlign paper is different from the analysis we conducted here. The analysis here focused on identifying pathways that are potentially strongly regulated by non-CN mechanisms that can overcome CN dosage effects. Whereas the CloneAlign analysis identifies differentially expressed pathways between clones that are not directly caused by CN dosage effects.

[C7]: The main novelty and benefit of TreeAlign reside mostly in the ability to identify gene dosage effects. Methodologically, this consists of adding one additional binary random variable to the Clonealign model, which is a simple extension. It is unclear whether the most reasonable way to model gene dosage is via a binary variable, instead of a continuous one reflecting the different sensitivities that different genes have to copy numbers.

[R7]: In TreeAlign, we encode whether a gene is dependent on CN dosage effects through the binary variable k and place a Dirichlet prior $p(k)$ on k . $p(k)$ is continuous and can reflect the different sensitivities that genes have to copy numbers to some extent. Mathematically k acts as a conditional switch which then allows for different likelihood terms to be used according to the condition. This is a standard approach in probabilistic graphical modeling. In addition, by conditioning on k , we still get a continuous estimation of CN dosage effects which can be interpreted as probability. Using the binary variable k also allows for easier future extensions of TreeAlign if we want to plug in different models for gene expression in the CN-dependent/independent scenarios. We also tested using a continuous k drawn from a uniform distribution to represent CN dosage effects previously and observed similar performance on simulated datasets as compared to the current setup. In the final implementation, we decided to proceed with a discrete variable k for its interpretability and flexibility.

[C8]: The allelic imbalance model is a useful addition, even though it does not make a substantial improvement to the cell-to-clone assignments. For the allelic imbalance, do they only care about cases with LOH and CN-neutral LOH, or do they consider other cases with allelic imbalance without LOH? E.g. 3 copies of one allele and 1 copy of another.

[R8]: Both LOH and allelic imbalance without LOH were considered by the B allele frequency (BAF) input of TreeAlign. BAF ranges between 0 and 1. BAF=0|1 represents LOH. BAF=0.5 represents a balanced copy number from 2 alleles. Other BAF represents allelic imbalance without LOH. More details on how we use BAF inferred from scDNA data in TreeAlign are now described in **Methods (Line 386)**.

[C9]: The allele-specific model is not described in detail, and only an overview is given. The authors must describe the complete Bayesian model. Additionally, they state their model is a Bayesian one, but do not show the prior distributions for the random variables in their model, or indeed even specify clearly which are random variables and which are learnable parameters. Is there a prior on the per-copy expression μ_{g0} ? And on the random effects part? How are they optimized? What are Delta distributions? The authors mention variational inference, but they also mention MAP estimation. Which do they use? What is the input to the allele-specific

model? Some of this information is scattered between Figure 1, Table 1, and the Methods section, with no explicit correspondence. The authors must provide a complete and detailed description of the method in the Methods section.

[R9]: We rewrote **Methods** to provide a more detailed description of model setup and inference (**Line 357-417**).

[C10]: In the model (Equation 3), the noise factors are only active for genes with no clonal effects. Perhaps they are missing a parenthesis to make sure all terms are affected?

[R10]: We thank the reviewer for catching this error. We have corrected the equation in **Methods** accordingly (**Line 373**).

[C11]: In the graphical model (Figure 1 c), what is the random variable y ? Where is its dependence on the parent nodes described?

[R11]: Variable y represents the expected expression of genes in scRNA. The dependencies of y were explained in the updated **Methods** (**Line 371-385**).

[C12]: Lines 555-556: The authors state that the total-CN and the allele-specific models were combined for the integrated TreeAlign model, but provide no details on how they were combined formally.

[R12]: The total CN model and allele-specific model share categorical variable z_n which indicates the clone assignment of cell n . The likelihood of the integrated model is the product of the likelihood of the total CN model multiplied by the likelihood of the allele-specific model. Therefore, z_n can be inferred from the two models separately or combined depending on the input data provided.

[C13]: How does the model deal with dosage of very high copy numbers? Presumably, a gene with 10 copies might have a lower than 5-fold increase in gene expression through some mechanism of regulatory compensation, as described in the Clonealign paper. Do the authors also truncate the input CNVs to their model?

[R13]: By default, TreeAlign truncates CNs > 10 to 10 and represents the CN-expression relationship with a linear function. If the expression of a gene is lower than expected based on CN changes, it will have lower $p(k)$ suggesting reduced CN dosage effects and potential regulatory compensation mechanisms. Functions that are more biologically meaningful could be used to replace the current setup in the future (**Discussion, Line 332**).

[C14]: The method may take as input a single-cell phylogenetic tree with copy numbers, which may be unreliable at the single-cell level due to the sparse nature of whole-genome scDNA-seq data, and the difficulty of building a tree from it. While this is somewhat bypassed through the

recursive fitting scheme employed by the authors, a discussion of the effect of the accuracy of the tree structure used as input should be included.

[R14]: We thank the reviewer for this important suggestion. To further investigate the influence of inaccurate phylogeny input on TreeAlign, we randomly selected different proportions of CN profiles from scDNA and shuffled their cell labels in patient 22. With more cell labels being shuffled, the tree will become less accurate in reflecting the true phylogeny of the population. When less than 20% of cells were shuffled, TreeAlign was able to resolve the same number of subclones as with the original data (**Extended Data Fig. 4**). With more than 20% but less than 50% cells shuffled, TreeAlign was able to assign cells to major subclades but failed to resolve more closely related subclones. When more than 50% cells were shuffled, TreeAlign failed and assigned all expression profiles to the unassigned state. These results suggest that inaccurate single cell phylogeny does affect TreeAlign performance, but partially inaccurate phylogeny can be tolerated to some extent.

Extended Data Fig. 4. **Clone assignment accuracy of TreeAlign with shuffled phylogenies.** **a**, Heat map of clone assignment in patient 022. Columns represent input phylogenies with certain % of cell labels being randomly shuffled. **b**, Adjusted rand index of clone assignment using shuffled phylogenies in patient 022. Clone assignment results with the original phylogeny were used as ground truth for comparison.

[C15]: Since the method involves multiple rounds of fitting a Bayesian model to data with different combinations of subclonal copy number profiles as input, the authors must report how many rounds of fitting they have to go through for each tree in the real data.

[R15]: For the PDX and cell line datasets, the integrated model finished with 1-10 rounds of fitting and the total CN model finished with 1-3 rounds of fitting when we ran TreeAlign with the phylogeny input (Extended Data Fig. 9 and 10). For patient 22, the integrated model was fitted for 9 rounds and the total CN model was fitted for 6 rounds.

Extended Data Fig. 9. **Inference of integrated TreeAlign in PDXs and cell lines. a**, rounds of fitting the integrated model with phylogeny input.

Extended Data Fig. 10. **Inference of total CN TreeAlign in PDXs and cell lines. a**, rounds of fitting the total CN model with phylogeny input.

[C16]: Can the iterations of fitting the Bayesian model deeper in the tree be initialized from the results from previous iterations at lower depths in the tree? This might significantly reduce computation time, instead of re-initializing the fit from scratch every time.

[R16]: At each iteration, TreeAlign will select genes with CN differences between subtrees as input so each iteration would have different sets of genes as input which makes it hard to initialize from previous iterations. However, when TreeAlign proceeds down a phylogeny, fewer genes and cells will be included in later iterations. This usually accelerates the inference as we noticed that the inference time of TreeAlign tends to be shorter with fewer genes and cells in input (**Extended Data Fig. 9e, f**). Run time of TreeAlign is generally acceptable on real cancer datasets. With better parallelization, we believe it could be further improved.

Extended Data Fig. 9. **Inference of integrated TreeAlign in PDXs and cell lines.** e, Scatter plot showing the time to finish for each run as a function of the number of cells in scRNA input. f, Scatter plot showing the time to finish for each run as a function of the number of genes in scRNA input.

[C17]: The authors state that alternative methods, including Clonealign, SCATrEx and CCNMF “require using predefined subclones from scDNA data as input”. On the one hand, this is not true for CCNMF, which takes as input single-cell copy number profiles, instead of subclones, and finds a clustering that matches both the scRNA and the scDNA data simultaneously. On the other hand, the TreeAlign model as described in the Methods section also takes predefined subclones from scDNA data. The only difference is in the pre-processing and post-processing that selects the appropriate subclones, by using simple criteria such as the number of cells in a subclone from the scDNA data, or the percentage of scRNA cells confidently assigned to subclones.

The use of the phylogenetic tree to define the depth at which clones should be defined for effective mapping between the two technologies can essentially be seen as a wrapper function around Clonealign that checks the quality of the model fit to stop increasing the subclonal granularity. This procedure can be easily applied to any method that performs the scRNA-scDNA mapping, including Clonealign, SCATrEx and, in principle, CCNMF.

We thus believe that the authors must rephrase their method overview to make this aspect explicit. For example, stating that “TreeAlign is a Bayesian probabilistic model that maps gene expression profiles from scRNA to phylogenies from scDNA which obviates the need to identify clones a priori from a tree” is not accurate. Rather, “TreeAlign is a software package that fits a Bayesian probabilistic model that maps gene expression profiles from scRNA to clones from scDNA that are refined recursively based on the quality of the model fit at each level of the scDNA phylogenetic tree” would be a fairer description. To be clear, we believe this procedure is a good idea, but its scope should be evident in the text.

[R17]: We thank the reviewer for this suggestion. We modified **Introduction and Results (Line 81 and Line 95)** accordingly to better account for the scope of TreeAlign.

Figure comments

[C18]: Figure 1: not all variables in the graphical model are described in the text: t , r , b , and π , for example.

[R18] We have updated **Methods and Extended Data Fig. 1** to include the description and prior distribution for all variables in the graphical model.

[C19]: Figure 2 b: it is confusing to have the nodes and the clones in the figure. Why not just label the clones? Cells are assigned to clones, right? Plus, in subsequent figures nodes are not labelled anymore, only clones.

[R19]: We have removed the node labels in **Fig. 2b** and **Extended Data Fig.3**.

[C20]: Figure 2 c: using percentages to denote expression levels is unclear. Perhaps the authors can make it more explicit what the facets mean in the figure.

[R20]: In addition to using percentages to denote expression levels, we added the ranges of normalized expression of genes in **Fig. 2c**. The normalized expression was calculated as implemented in the `NormalizeData` function in the R package `Seurat` where features counts are divided by the total counts for that cell and multiplied by the `scale.factor=10000` followed by natural-log transformation.

[C21]: Figure 3 d and Figure 4 a: please clarify the meaning of the colors in the pie charts. What does blue encode, if there is no blue clone?

[R21]: The colors in the pie charts represent intermediate subtrees or final clones that cells in scRNA were assigned to during each iteration of clone assignment by `TreeAlign`. For example, in **Fig. 3d**, the leftmost pie chart represents the proportions of cells assigned to the two main subtrees. The light green color of the outer ring represents the root of the phylogeny. The red color represents the subtree on the top or clone A. the blue color represents the bottom subtree which encompasses clone B, C and D. We added additional description to the caption of **Fig. 3d** to better explain the color annotation.

[C22]: Figure 4f: How do they compute the AUC? What's the ground truth?

We used the variable a estimated from scDNA data by `SIGNALS` as ground truth. The binary variable a to indicate whether the reference allele at a heterozygous SNP should be assigned as B allele. a can be obtained using `SIGNALS` which uses information from scDNA to phase the SNPs in scRNA and assign alleles accordingly. We can also treat a as a hidden variable and jointly infer it from `TreeAlign`. Comparing a inferred from `TreeAlign` to the `SIGNALS` output allows us to estimate the performance of `TreeAlign`. We updated **Methods (Line 397)** to include more descriptions of the random variable a .

[C23]: Figure 5 a: Please label the rows and columns of the heatmaps.

[R23]: We updated **Fig. 5a** to include labels for rows and columns.

[C24]: Figure 6 a: The caption reads that the rows are genes and the columns are clones, but the rows are also coloured by clones. Why?

[R25]: The rows represent genes that are upregulated in each clone. The row annotation color represents the clone in which the genes are upregulated. To avoid confusion, we removed the row color annotations.

[C25]: Figure 6 b, c, and d: the caption labels do not match the subfigures. Example: the legend says b-c contains proportions of DE genes, but b contains a UMAP and c contains a volcano plot. This also happens in Figure 9.

[R25]: We have corrected the mismatch in **Fig. 6**.

Referee #2

The manuscript by Shi, et al solves the unmet need to link subclonal CNAs to clone-specific phenotypes using single cell sequencing. As some cancers are driven in large part by CNA (versus somatic mutations), this is an important area of study. The method can also improve subclonal assignments as well as enabling the inference of genes that are mechanistically altered through subclone specific CNAs, and identification of expression programs that are genomically independent. This tool is really quite clever. One challenge for application of the method is that most datasets, especially patient samples, lack both scRNAseq and scDNAseq data, and more frequently combine bulk DNAseq with scRNAseq, for example. However, perhaps this is a first step towards leveraging alternative data types. I only have some minor to moderate suggestions, mostly aimed at clarifying functionality and details of the method. Overall, this is an important new and original approach. The study could be strengthened by providing a broader summary of results across patients/datasets as a whole (instead of highlighting specific samples) and a more in depth description of the method.

Comments

[C1]: For the patient and cell line data, it would be helpful to assess the relationships between the inferCNV (or copykat) CNVs with the TreeAlign method as a whole across all patients/etc (both advantages, which there clearly are, as well as any results the detail performance of the method when applied to different data types/quality).

[R1]: We applied InferCNV on the patient, PDX and cell line data, and assigned expression profiles to genomic clones based on pearson correlation coefficients between CN from scDNA and InferCNV-corrected expression profiles. We assessed the clonal frequencies estimated by InferCNV and TreeAlign (**Extended Data Fig. 11**). Both are significantly correlated with clonal frequencies from scDNA suggesting that both methods can capture copy number signals from

scRNA data. However, Clonal frequencies estimated by TreeAlign in scRNA are more concordant with DNA data suggesting TreeAlign performs better in terms of clone assignment compared to InferCNV. It should also be noted that the overall performance of both InferCNV and TreeAlign in the PDX/cell line dataset is worse compared to the patient 22 dataset, probably due to fewer CN differences between clones and more homogenous cell populations in the PDX/cell line dataset.

Extended Data Fig. 11. **Compare InferCNV and TreeAlign subclone frequencies. a-b,** Correlation between clone frequencies estimated by scRNA-data (x axis) and scDNA-data (y axis) by TreeAlign and InferCNV in (a) HSGC PDXs and cell lines and (b) patient 22

[C2]: Figure 3D and 4A are beautiful and really highlight the strength of the method. However, is this result typical across patients? It would be helpful if the additional patient results are included as supplemental files/data. When does the approach fail (low read count, minimum number of cells, etc)? What parameters are important for linking data effectively? In general, the methods (and results) could be expanded on and more detailed.

[R2]: We included additional supplementary files for the PDX/cell line dataset. The integrated model of TreeAlign generally performs well in this dataset and only failed for cell line SA906a due to a low number of genes ($n=32$) with CN differences and heterozygous SNPs ($n=7$) with BAF differences between subclones.

By subsampling experiments of patient 022, we noticed that decreasing the number of input genes compromised the performance of TreeAlign (**Fig. 4g**), suggesting that TreeAlign has worse performance on datasets that are more homogeneous in terms of CNAs. However, the method is robust to decreasing numbers of cells in scRNA. With only 5% of cells ($n=206$) of the original patient 22 scRNA dataset, the adjusted rand index of clone assignments were close to 1 suggesting highly concordant results in comparison to the full dataset.

With additional simulations, we also noticed that inaccurate tree construction could influence TreeAlign's performance. When less than 20% of cells were misplaced on phylogeny, TreeAlign was able to resolve the same number of subclones as with the original data. When more than

50% cells were misplaced, TreeAlign failed and assigned all expression profiles to the unassigned state.

We included additional limitations of TreeAlign in **Discussion (Line 328)**.

Fig. 4. **Incorporating allele specific expression increases clone assignment resolution.** g, Robustness of clone assignment to gene subsampling in patient 022. Adjusted rand index was calculated by comparing clone assignments from subsampled datasets to the complete dataset.

[C3]: In addition, both of these datasets include relatively small numbers of cells--are there larger cohorts that could also be example from additional CNA rich tumors to provide a broader survey of the method's applicability, advantages and weaknesses?

[R3]: Unfortunately, publicly available matched multimodal datasets of scDNA and scRNA are still limited. In addition to the two datasets, we also applied TreeAlign on data from the gastric cancer cell line NCI-N87 (675 cells in scDNA and 3212 cells in scRNA). TreeAlign was able to assign expression profiles to 3 subclones identified in scDNA. The clonal frequencies estimated from scRNA were also highly concordant with scDNA. Also note that different from the other two datasets, scDNA data here was generated with 10x genomics single-cell CNV instead of DLP+. It demonstrates that TreeAlign also works for 10x scDNA data.

[C4]: The phenotype examples are somewhat peripheral to the study and it is currently hard to assess the accuracy of the TreeAlign approach compared to others. Is there a way to calculate the phenotypic diversity within and across subclonal populations using the TreeAlign versus InferCNV subclonal calls? The hypothesis would be that the cells from each subclone have more similar phenotypes to each other than to cells from other subclones.

[R4]: To evaluate clone-level transcriptional heterogeneity, we calculated Pearson correlation coefficients and Euclidean distance between cells using the top 20 principal components of the gene expression matrices. We found that cells sampled from the same TreeAlign clone or InferCNV clone tend to have higher correlation and lower distance between them (**Extended Data Fig. 12**), suggesting lower transcriptional diversity within the subclonal populations.

Extended Data Fig. 12. **Subclonal transcriptional diversity.** **c-d**, mean Euclidean distance between cells in scRNA-data sampled across or within subclones for (c) HSGC PDXs and cell lines and (d) patient 022. **e-f**, mean Pearson correlation coefficient between cells in scRNA-data sampled across or within subclones for (e) HSGC PDXs and cell lines and (f) patient 022.

References

1. Martins, F. C. *et al.* Clonal somatic copy number altered driver events inform drug sensitivity in high-grade serous ovarian cancer. *Nat. Commun.* **13**, 6360 (2022).
2. Bhattacharya, A. *et al.* Transcriptional effects of copy number alterations in a large set of human cancers. *Nat. Commun.* **11**, 715 (2020).

REVIEWERS' COMMENTS

Reviewer #1 (Remarks to the Author):

The authors have addressed all our concerns. They added more simulations, commented on the real data results, and significantly improved the model description. While we still believe that novelty is limited, the method achieves an improvement over the state of the art in the cell-to-clone assignment task and provides dosage estimates that may be useful. We have a few remaining comments.

Why does the model require two per-copy expressions for each gene depending on whether the gene is dosage-sensitive? Shouldn't the per-copy expression be independent of this, and thus require only one parameter? We would think that the most natural way of modelling binary gene dosage effects would be as $\mu_g \frac{\lambda_{cg}^2}{k_g}$. Can the authors clarify their choice?

We appreciate that the authors added a simulation study to evaluate the impact of using allelic-specific information in their model. The results show that for dosage effects present in 70% of genes, which is a case the authors argue is plausible for real data, for 1000 genes (unclear what this number means exactly – see next comment), using SNPs does not help the clone assignment accuracy. For 100 genes, the effect is barely noticeable. For dosage effects in 30% and 50% of genes, the effect is clear. The conclusion seems to be that using dosage effects is the main driver of the performance of TreeAlign, and that using allele-specific signal is a possibly useful addition in edge cases. For completeness, the authors must place the real data numbers in the context of this simulation study, indicating for example, how many SNPs were identified in the patient O22 data in Section 2.4.

Extended Data Fig 5: What is the number of genes facet denoting? Is it the number of genes affected by copy number on the scDNA data? The number of genes on which different clones have different copy numbers? Please clarify.

The fact that the allele-specific information adds little to the model performance seems to make the title of this manuscript quite inappropriate. Instead, it seems that TreeAlign mostly exploits varying gene dosage effects to perform the genotype-phenotype mapping in cancer cell populations. We encourage the authors to reconsider the title.

The authors state in their rebuttal that “ $p(k)$ is continuous and can reflect the different sensitivities that genes have to copy numbers to some extent”. We disagree with this statement as $p(k)$ does not interact with the likelihood for the gene expression data directly, but rather indirectly through k , which is binary. The authors should add a clear motivation for modelling gene dosage as a binary effect instead of a continuous one to the text.

Extended Data Fig 9 and 10: what does TreeAlign “with phylogeny input” mean? Doesn't TreeAlign always use the phylogeny as input (isn't this why it's called TreeAlign)? And why is the patient data not included in these figures?

In the new model description, λ represents the copy number values, but the way it goes into the

likelihood would indicate that it is actually the copy number values divided by 2.

Line 364: typo in Dirichlet

Reviewer #2 (Remarks to the Author):

The authors have addressed review comments. I believe their research is impactful and high quality. I recommend publication.

Reply to reviewer reports on Nature Communications submission of the manuscript:

“Exploiting allele-specific transcriptional effects of subclonal copy number alterations for genotype-phenotype mapping in cancer cell populations”

Title now updated to “Allele-specific transcriptional effects of subclonal copy number alterations enable genotype-phenotype mapping in cancer cells”

We would like to thank the reviewers for careful consideration of our manuscript NCOMMS-23-02664-T, now titled “***Allele-specific transcriptional effects of subclonal copy number alterations enable genotype-phenotype mapping in cancer cells***”. We have revised the manuscript to address reviewer #1’s comments and editorial requests. Detailed changes to our manuscript and point-by-point responses to the comments of the reviewers are included below.

Referees' comments:

[C=comment, R=reply]

The authors have addressed all our concerns. They added more simulations, commented on the real data results, and significantly improved the model description. While we still believe that novelty is limited, the method achieves an improvement over the state of the art in the cell-to-clone assignment task and provides dosage estimates that may be useful. We have a few remaining comments.

[C1]: Why does the model require two per-copy expressions for each gene depending on whether the gene is dosage-sensitive? Shouldn't the per-copy expression be independent of this, and thus require only one parameter? We would think that the most natural way of modelling binary gene dosage effects would be as $\mu_g \frac{\lambda_{cg}}{2}^{k_g}$. Can the authors clarify their choice?

[R1]:

To Clarify, μ_{g1} does not represent the per-copy expression but represents the base expression of a gene if its expression is independent of CN dosage effects. We updated Method (line 370) and Figure S1 to further clarify the definition of μ_{g0} and μ_{g1} .

Our modeling makes the assumption that the expression of a gene can take on 2 modes: 1), dependent on CN. 2), independent of CN. Hence, the model is similar to a mixture model where data points (gene expression of cells) are assumed to be from different distributions based on dosage effect dependencies. We think this setup is more general, flexible and interpretable in terms of Bayesian probabilistic modeling. Though currently, for genes whose expression is dependent on CN, we model their expression using a simple linear model, our framework based on the binary parameter k and the conditional probability modeling would allow investigators to easily test more complex gene dosage effects in future work by changing the function where expected expression is constructed.

It is unclear whether modeling of gene expression as $\mu_g \frac{\lambda_{cg}}{2}^{k_g}$ (as proposed by Reviewer #1) is a natural setup as it also assumes an exponential relation between gene expression and k , which is another arbitrary choice and may not represent the true distribution of dosage effects.

We appreciate that the authors added a simulation study to evaluate the impact of using allelic-specific information in their model. The results show that for dosage effects present in 70% of genes, which is a case

the authors argue is plausible for real data, for 1000 genes (unclear what this number means exactly – see next comment), using SNPs does not help the clone assignment accuracy. For 100 genes, the effect is barely noticeable. For dosage effects in 30% and 50% of genes, the effect is clear. The conclusion seems to be that using dosage effects is the main driver of the performance of TreeAlign, and that using allele-specific signal is a possibly useful addition in edge cases. For completeness, the authors must place the real data numbers in the context of this simulation study, indicating for example, how many SNPs were identified in the patient 022 data in Section 2.4.

Supplementary Figure. 5. **Clone assignment accuracy with simulated allelic data.** Accuracy of clone assignment for the integrated model of TreeAlign on simulated scRNA datasets as a function of varying numbers of heterozygous SNPs in input. Panels represent datasets with different numbers of genes in clone-specific copy number regions (genes with copy number differences between clones) and proportions of genes with CN dosage effects.

[R]: When dosage effects are present in 70% of genes, including allele specific information (e.g. adding 1000 heterozygous SNPs in TreeAlign input) significantly improved clone assignment accuracy in the datasets with 100 and 500 input genes. For simulated datasets with

1000 genes, the difference is too small to notice mainly due to the model performance being too good on relatively large simulated datasets. With real datasets, allele-specific information is more important in distinguishing clones that are closely related and highly similar. For example, patient 022 has 6538 heterozygous SNPs that can be used for TreeAlign input. These SNPs provided critical information in distinguishing clone D.4 from D.2 and D.3 which only have 355 genes with different copy numbers between them. For completeness, we included the number of SNPs that can be used as TreeAlign input for patient 022 and cell line/PDX samples from Funnell et al. in Figure S9.

Supplementary Figure. 9. **Number of heterozygous SNPs used as input for TreeAlign in samples from Funnell et al. and patient 022.**

[C]: Extended Data Fig 5: What is the number of genes facet denoting? Is it the number of genes affected by copy number on the scDNA data? The number of genes on which different clones have different copy numbers? Please clarify.

[R]: To clarify, it is the number of genes with different CNs between clones, or genes located in clone-specific copy number (CSCN) regions. We updated the caption of Figure S5 to make it clearer.

[C]: The fact that the allele-specific information adds little to the model performance seems to make the title of this manuscript quite inappropriate. Instead, it seems that TreeAlign mostly exploits varying gene dosage effects to perform the genotype-phenotype mapping in cancer cell populations. We encourage the authors to reconsider the title.

In simulation experiments using a large number of genes (e.g. 1000 genes with CN differences between clones) as input, TreeAlign can achieve near perfect performance which makes it impossible to discern the contribution of allele-specific information. However, with real data, especially when TreeAlign proceeds further down a phylogenetic tree, there will be fewer genes with copy number differences between subclones. In these cases, the allele-specific information contributed to clone assignment and allowed for better characterization of subclones as shown in Figure 4. Furthermore, we have demonstrated that the integrated model outperforms all others and should be used in practice. Therefore, we believe it is critical to include allele-specific information so that TreeAlign would have much better performance on real cancer data. We updated the title to “Allele-specific transcriptional effects of subclonal copy number alterations enable genotype-phenotype mapping in cancer cells”.

[C]: The authors state in their rebuttal that “ $p(k)$ is continuous and can reflect the different sensitivities that genes have to copy numbers to some extent”. We disagree with this statement as $p(k)$ does not interact with the likelihood for the gene expression data directly, but rather indirectly through k , which is binary. The authors should add a clear motivation for modelling gene dosage as a binary effect instead of a continuous one to the text.

[R]: We added additional text to Line 94 and Line 322 to better describe the motivation for modeling the presence of gene dosage effects as a binary variable. In brief, encoding the presence of dosage effects as a binary indicator allows us to frame the problem as a conditional probability distribution which separates the expected expression into two components: 1), dependent on dosage effects 2), independent of dosage effects. It makes TreeAlign a flexible framework as it would be easy to change how expected expression is modeled in both conditions and explore more complex patterns of transcriptional regulations in the future. With this statistical framework, another extension could be to model k as a categorical variable with a 1 of K Multinomial distribution where the different components represent different functional forms of gene dosage (e.g. linear vs logistic vs exponential, etc...).

[C]: Extended Data Fig 9 and 10: what does TreeAlign “with phylogeny input” mean? Doesn’t TreeAlign always use the phylogeny as input (isn’t this why it’s called TreeAlign)? And why is the patient data not included in these figures?

[R]: The TreeAlign Python package can take in both phylogenies or pre-defined clones as input depending on the user's choice. If pre-defined clones are provided as input, TreeAlign will not conduct the recursive assignment process but assign expression profiles to the predefined clones directly.

The extended data Fig 9 and 10 only included data from Funnell et al. The patient data (patient 022) are from a different cohort, therefore we did not include them here to compare with the cell line and PDX data. For the integrated model, it took an average of 314.4 iterations and 247.074 seconds to reach convergence in patient 022. For the total CN model, it took an average of 236.5 iterations and 79.204 seconds to reach convergence in patient 022.

[C]: In the new model description, λ represents the copy number values, but the way it goes into the likelihood would indicate that it is actually the copy number values divided by 2.

[R]: λ represents the copy number when gene expression is dependent on dosage effects.

Line 364: typo in Dirichlet

[R]: Thank you. We have corrected the typo.